DOI: 10.1038/s41467-017-02023-z　　OPEN

# Alternative activation generates IL-10 producing type 2 innate lymphoid cells

Corey R. Seehus[1], Asha Kadavallore[1], Brian de la Torre[1], Alyson R. Yeckes[1], Yizhou Wang[2], Jie Tang [2] & Jonathan Kaye [1,3]

Type 2 innate lymphoid cells (ILC2) share cytokine and transcription factor expression with $CD4^+$ $T_h2$ cells, but functional diversity of the ILC2 lineage has yet to be fully explored. Here, we show induction of a molecularly distinct subset of activated lung ILC2, termed $ILC2_{10}$. These cells produce IL-10 and downregulate some pro-inflammatory genes. Signals that generate $ILC2_{10}$ are distinct from those that induce IL-13 production, and gene expression data indicate that an alternative activation pathway leads to the generation of $ILC2_{10}$. In vivo, IL-2 enhances $ILC2_{10}$ generation and is associated with decreased eosinophil recruitment to the lung. Unlike most activated ILC2, the $ILC2_{10}$ population contracts after cessation of stimulation in vivo, with maintenance of a subset that can be recalled by restimulation, analogous to T-cell effector cell and memory cell generation. These data demonstrate the generation of a previously unappreciated IL-10 producing ILC2 effector cell population.

[1] Research Division of Immunology, Departments of Biomedical Sciences and Medicine, Samuel Oschin Comprehensive Cancer Institute, Cedars-Sinai Medical Center, 8700 Beverly Blvd., Los Angeles, CA 90048, USA. [2] Genomics Core Facility, Cedars-Sinai Medical Center, 8723 Alden Drive, Los Angeles, CA 90048, USA. [3] Department of Medicine, David Geffen School of Medicine, University of California, Los Angeles, 10833 Le Conte Ave., Los Angeles, CA 90095, USA. Correspondence and requests for materials should be addressed to J.K. (email: kayej@csmc.edu)

The immune system utilizes a diverse array of cell subtypes that can eradicate pathogens efficiently, while also repressing autoimmunity. Cells of the innate immune system termed innate lymphoid cells (ILC), have been identified in mice and humans, and helper-like ILC have many parallels to CD4[+] helper T (T$_h$) effector cell subsets[1], despite a lack of antigen receptors. In this regard, some subsets within the type 1 ILC (ILC1), ILC2, and type 3 (ILC3) populations have been compared to T$_h$1, T$_h$2, and T$_h$17 cells, respectively. Both T$_h$2 cells and ILC2 secrete the cytokines IL-5 and IL-13, are dependent on the transcriptional regulator GATA-3, and express similar regulomes in response to infection[2]. ILC2 have a beneficial role in eradication of parasitic helminths[3], restoration of lung epithelial barrier function following influenza infection[4], and regulation of beige fat biogenesis[5]. Although ILC2 elicit beneficial host responses to pathogens and mucosal damage, these cells are also implicated in disease, most notably allergic responses in the lung[6].

Subpopulations of T$_h$ effector cells arise during activation of mature naïve CD4[+] T cells as a consequence of distinct environmental cues, thereby yielding highly adaptable responses. By contrast, ILC subtypes arise from a common immature bone marrow precursor in a developmental program[7], and thus specific effector cell differentiation was thought to be less influenced by external signals. However, data now show that plasticity exists within ILC3 and ILC2, primarily driven by induction of T-bet and development of an ILC1-like effector

**Fig. 1** In vivo activation of lung ILC2 induces *Il10* expression. **a** Flow cytometry analysis of ILC2 from the lungs of wildtype animals treated with IL-33 (right) or PBS (left). The frequency of ILC2 (Lin⁻ST2⁺) within the CD45⁺Thy-1.2⁺ cell population is indicated. **b**, **c** Frequency (**b**) and number (**c**) of lung ILC2, calculated from gates as in **a**, here and in all subsequent figures. **d** Volcano plot comparison of whole transcriptome gene expression of ILC2 from IL-33 and PBS-treated animals. Differentially expressed genes (defined as statistically significant adjusted *P* < 0.05) with changes of at least twofold are shown in red (Student's *t* test with Benjamini and Hochberg correction). The top ten differentially expressed genes ranked by fold-change in either direction are labeled. **e**–**g** Selected genes plotted as the average of the fragments per kilobase of transcript per million mapped reads (FPKM) in IL-33-treated vs. PBS-treated animals. Differentially expressed genes (defined as in **d**) are labeled in red. Gray area indicates region of twofold or lower change in expression. Shown are expression of selected genes grouped by (**e**) ILC2 markers and markers of activation, (**f**) transcription factors, and (**g**) effector molecules. Data are representative of four independent experiments with one mouse used per experiment **a**–**c**, or are from three experiments with cells pooled from five mice each **d**–**g**. Each symbol **b**, **c** represents an individual mouse and horizontal lines indicate the mean here, and in all subsequent figures. ***P < 0.001 (Student's *t* test)

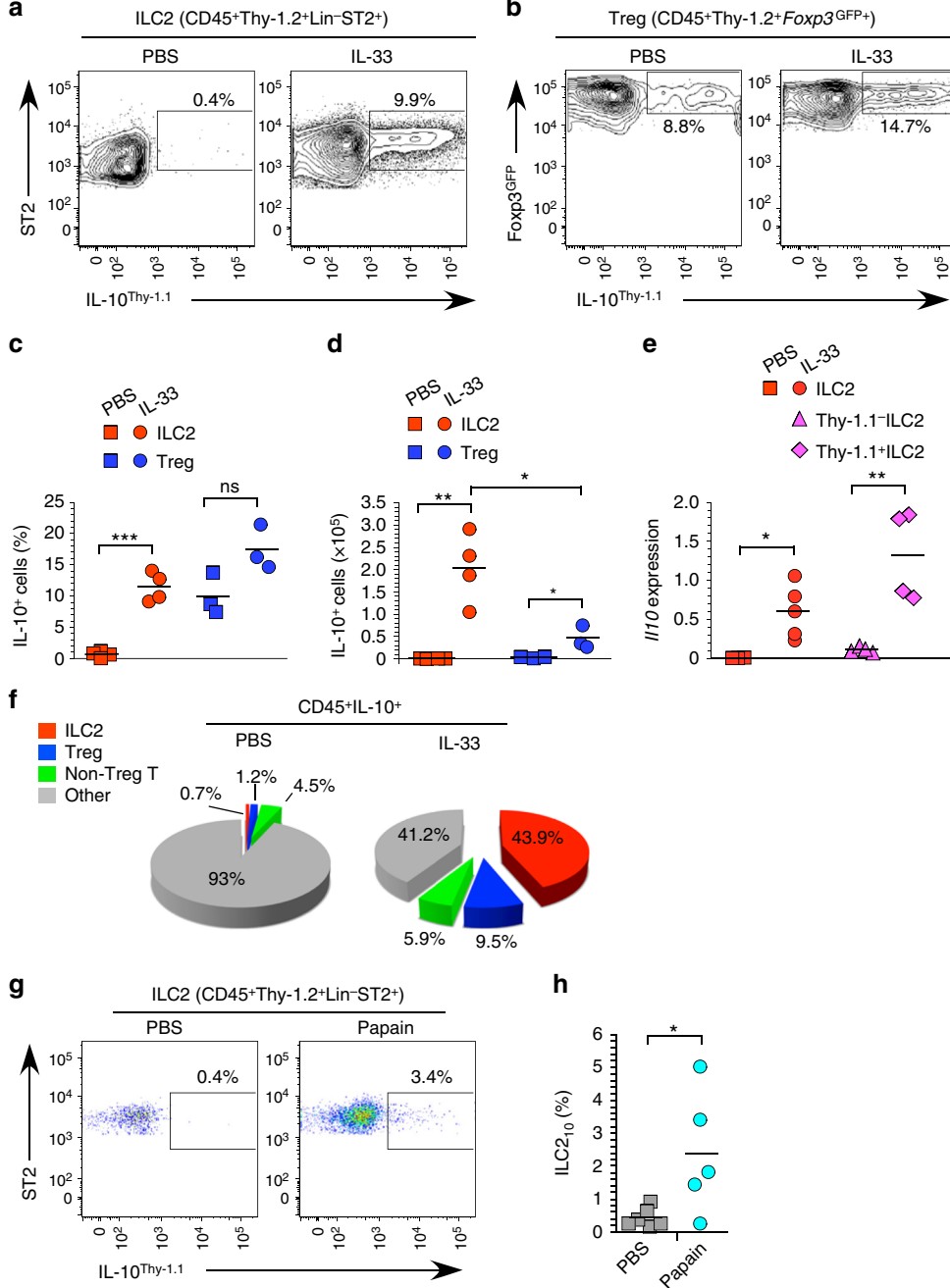

**Fig. 2** ILC2$_{10}$ constitute a major subset of IL-10 producing lung hematopoietic cells. **a**, **b** Flow cytometry analysis of ILC2 (**a**) or T$_{reg}$ cells (**b**) from the lungs of 10BiT*Foxp3*$^{GFP}$ dual reporter mice treated with PBS (left) or IL-33 (right). **c**, **d** Quantitation of the frequency (**c**) and number (**d**) of lung ILC2$_{10}$ and T$_{reg}$ cells, calculated from gates as in **a** and **b**, here and in all subsequent figures. **e** *Il10* gene expression from total ILC2 from wild-type mice (red) or subsets of ILC2 from 10BiT*Foxp3*$^{GFP}$ mice (pink) treated with PBS or IL-33. **f** Frequency of cell types including ILC2 as in **a**, T$_{reg}$ cells as in **b**, non-T$_{reg}$ T cells (Lin$^+$Thy-1.2$^+$GFP$^-$) or other cells (Lin$^+$Thy-1.2$^-$GFP$^-$), among the total CD45$^+$IL-10$^+$ population from the lungs of 10BiT*Foxp3*$^{GFP}$ animals treated with PBS or IL-33. **g** Analysis of lung ILC2$_{10}$ in 10BiT*Foxp3*$^{GFP}$ animals following chronic treatment with PBS or papain. **h** Compiled data of frequency of ILC2$_{10}$ in the lungs of papain-treated 10BiT*Foxp3*$^{GFP}$ mice gated as in **a**. Data are from at least two independent experiments with representative plots shown in **a**, **b**, **f**, **g** and compiled data on individual mice shown in **c**, **d**, **e**, **h**. ***$P < 0.001$, **$P < 0.01$, *$P < 0.05$, and ns (not significant, $P \geq 0.05$) (Student's *t* test)

program under inflammatory conditions[8, 9]. Whether external stimuli can also induce differential effector cell differentiation of ILC2, other than T-bet-dependent conversion to an ILC1-like cell, is unknown.

Here, we identify distinct IL-10 producing ILC2 effector cells, termed ILC2$_{10}$, that are induced by IL-33 and acquire an alternative activation phenotype. The ILC2$_{10}$ population undergoes contraction upon removal of stimulus, and can be recalled with subsequent challenge. In addition, these cells decrease expression of some genes associated with inflammation, and when induced in vivo, are associated with a decrease in eosinophil recruitment to the lung. ILC2$_{10}$ can also be induced by chronic exposure to the allergen papain, with the extent of induction correlating with the degree of activation of ILC2 and the inflammatory response. Together, these data identify ILC2$_{10}$ as a distinct effector cell population with immunoregulatory potential.

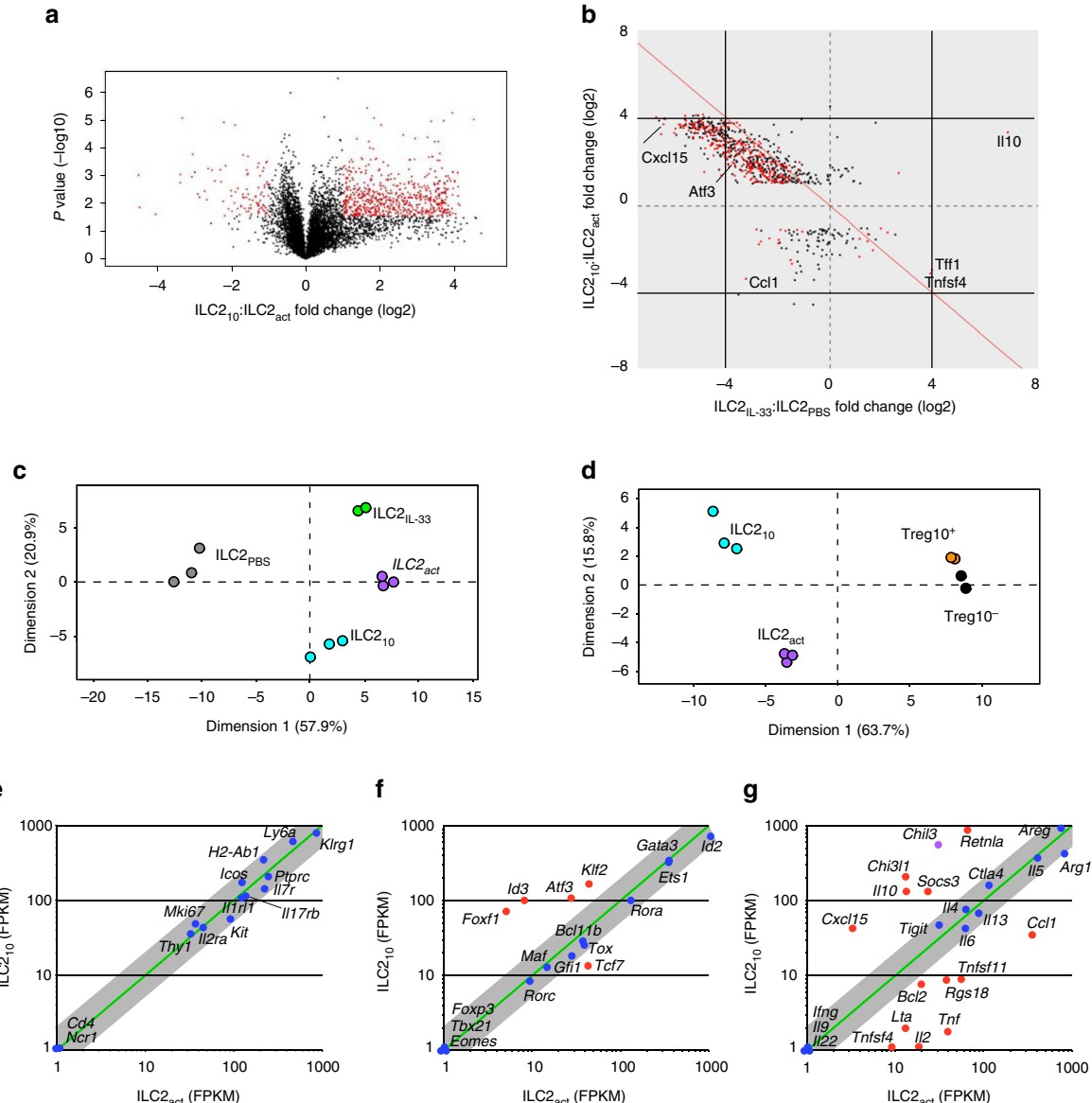

**Fig. 3** ILC2$_{10}$ represent a molecularly distinct ILC2 activation state. **a** Volcano plot for differentially expressed genes, defined as in Fig. 1, comparing ILC2$_{10}$ and IL-33 activated IL-10$^-$ ILC2 (ILC2$_{act}$). Genes defined as statistically significant analyzed as in Fig. 1 are shown in red. **b** Fold-change plot of differentially expressed genes in the intersection of indicated ILC2 populations. Genes defined as statistically significant in **a** are shown in red. **c** Principal component analysis of RNA-seq data from purified total ILC2 from PBS (ILC2$_{PBS}$) or IL-33 (ILC2$_{IL-33}$) injected mice, or from isolated ILC2$_{10}$ or ILC2$_{act}$ from IL-33 injected mice. **d** Principal component analysis of RNA-seq data comparing ILC2$_{10}$ and ILC2$_{act}$, with IL-10$^+$ and IL-10$^-$ T$_{reg}$ cells. **e–g** Selected genes plotted as the average FPKM from ILC2$_{10}$ vs. ILC2$_{act}$, presented as in Fig. 1. Shown are expression of selected genes grouped by (**e**) ILC2 markers and markers of activation, (**f**) transcription factors, and (**g**) effector molecules (*Chil3*, purple, $P = 0.05$). Data are from two or three independent experiments with five mice pooled per experiment **a–g**

## Results

**IL-33 or papain induces IL-10 producing ILC2.** We reasoned that a strong activation signal would reveal unknown ILC2 effector cell subpopulations. To test this, we injected mice with four daily doses of IL-33, a potent inducer of ILC2[10]. IL-33 injection resulted in significant expansion of ILC2 in the lung (Fig. 1a–c). To identify gene expression changes associated with IL-33-induced ILC2 activation, we performed RNA-seq on sorted lung ILC2 from mice injected with either vehicle or IL-33. Significant changes in gene expression, including both up- and downregulated genes were detected (Fig. 1d, Supplementary Data 1). Genes encoding cell surface molecules used for cell isolation (*Ptprc*, *Thy1*, and *Il1rl1*) or markers of the ILC2 subtype

(*H2-Ab1*, *Icos*, *Il2ra*) were expressed similarly in both cell populations, whereas genes encoding lineage markers (*Cd4*, *Cd3e*, *Ncr1*) were not expressed, confirming the identity of the isolated ILC2 (Fig. 1e). IL-33-mediated ILC2 activation led to other significant changes in gene expression (Fig. 1d), including upregulation of *Klrg1* and *Mki67*, encoding cell activation and proliferation markers (Fig. 1e), and *Il13*, *Il6*, and *Arg1* (Fig. 1g), involved in proliferation and inflammatory functions of ILC2[11]. Genes encoding transcriptional regulators associated with ILC2 development and/or function (*Id2*[12], *Gata3*[10], *Rora*[13], *Tcf7*[14], *Tox*[15], *Bcl11b*[16], and *Gfi1*[17]), were minimally or not differentially expressed between the cell populations (Fig. 1f). In contrast, *Tbx21* (encoding T-bet) was not expressed upon activation

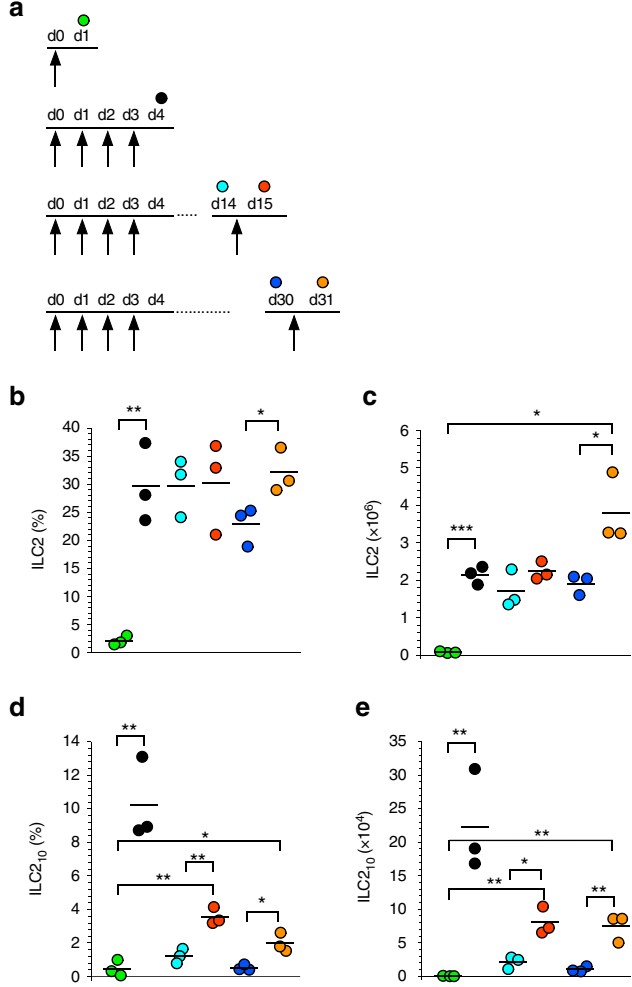

**Fig. 4** ILC2$_{10}$ effector cells exhibit memory-like properties in vivo. **a** Timeline for ILC2$_{10}$ recall experiment using 10BiT reporter mice, with colors corresponding to day of cell analysis. Arrows indicate IL-33 injections. **b**–**e** Frequency (**b**, **d**) and number (**c**, **e**) of total ILC2 (**b**, **c**) or ILC2$_{10}$ (**d**, **e**) with experimental protocol in **a**. Data are from one experiment with three mice per group (**b**–**e**). ***$P < 0.001$, **$P < 0.01$, *$P < 0.05$ (Student's $t$ test)

(Fig. 1f), nor was *Ifng* (Fig. 1g), indicating failure to convert to an ILC1-like gene program. Interestingly, *Foxp3*, the transcriptional regulator required for regulatory T (T$_{reg}$) cell differentiation[18], was not expressed (Fig. 1f), but genes encoding coinhibitory molecules TIGIT and CTLA4, and the anti-inflammatory cytokine IL-10 were upregulated in activated ILC2 (Fig. 1g). Inflammatory CD4$^+$ T cells that do not express FOXP3 but produce IL-9 and IL-10 have been identified[19]. We did not detect expression of *Il9* mRNA (Fig. 1g) in activated ILC2 and this cell population was negative for surface expression of CD4 and *Cd4* mRNA (Fig. 1e), demonstrating no contamination with this cell type.

IL-10 is a potent immunosuppressive cytokine that is expressed by distinct effector cell subsets of the immune system[20]. To determine if expression of *Il10* was limited to an ILC2 subset, we utilized a dual reporter mouse strain that contains the coding sequence of Thy-1.1 inserted into a BAC containing the *Il10* gene (10BiT)[21], and an IRES-GFP cassette knocked into the *Foxp3* locus (*Foxp3*$^{GFP}$)[22]. Using these mice, we identified subsets of ILC2 and T$_{reg}$ cells that expressed the *Il10* reporter in IL-33-treated animals (Fig. 2a–d). The fidelity of the reporter was confirmed, as *Il10* gene expression was highly enriched in Thy-1.1

$^+$ as compared to Thy-1.1$^-$ ILC2 (Fig. 2e). We hereafter refer to Thy-1.1$^+$ ILC2 as ILC2$_{10}$. T$_{reg}$ cells that expressed *Il10* modestly expanded in the presence of IL-33 (Fig. 2c, d), consistent with the presence of IL-33 responsive population of lung resident IL-10$^+$ effector T$_{reg}$ cells[23]. In 10BiT*Foxp3*$^{GFP}$ mice injected with PBS, over 90% of CD45$^+$ IL-10 producing cells in the hematopoietic compartment were non-T cells (Fig. 2f), likely macrophages[24]. However, in IL-33 injected mice, ILC2$_{10}$ accounted for, on average, 44% of the total CD45$^+$IL-10$^+$ population, equaling or surpassing the non-T-cell compartment (Fig. 2f). In contrast, *Foxp3*$^{GFP+}$ T$_{reg}$ cells and *Foxp3*$^{GFP-}$ T cells accounted for only 10% and 6% of the CD45$^+$IL-10$^+$ cell population, respectively. Thus, ILC2$_{10}$ represent a major IL-10-producing subpopulation of ILC2 that are induced by IL-33 in vivo.

The ILC2 population expands upon treatment with the protease allergen papain, and these cells orchestrate T$_h$2-driven immunity in the lung[25]. We have been unable to induce ILC2$_{10}$ in response to acute treatment with papain. However, a more chronic stimulation regimen induced ILC2$_{10}$ (Fig. 2g). There was significant variability in induction of ILC2$_{10}$ in individual animals (Fig. 2h). Notably, there was a strong positive correlation of ILC2$_{10}$ formation with the extent of inflammation and ILC2 activation, as assessed by increases in the frequency of tissue eosinophils and KLRG1 expression by ILC2, respectively (Supplementary Fig. 1). These results indicate that allergic responses can elicit ILC2$_{10}$ generation, and suggest that ILC2$_{10}$ formation is favored by chronic or highly inflammatory conditions.

**ILC2$_{10}$ are generated by alternative activation.** To determine if ILC2$_{10}$ are a molecularly unique subset of ILC2, we sort-purified Thy-1.1$^+$ and Thy-1.1$^-$ ILC2 from IL-33 injected reporter animals and performed RNA-seq analysis. In total, 774 genes were differentially expressed between ILC2$_{10}$ and IL-33-activated IL-10$^-$ ILC2 (ILC2$_{act}$) (Supplementary Data 2). Interestingly, the distribution of differentially expressed genes was skewed, with the vast majority more highly expressed by ILC2$_{10}$ than ILC2$_{act}$ (Fig. 3a). However, the majority of seemingly upregulated genes in ILC2$_{10}$, were instead genes that failed to downregulate upon activation with IL-33, with *Il10* a notable exception (Fig. 3b). Principal component analysis indicated that most of the variance between samples could be attributed to differences between experimental groups, suggesting biologically relevant differences (Fig. 3c). ILC2$_{10}$ and ILC2$_{act}$ were more transcriptionally distinct cell populations than were IL-10$^+$ and IL-10$^-$ splenic T$_{reg}$ cells isolated from identically treated reporter mice (Fig. 3d).

Genes encoding a number of transcriptional regulators and cell surface proteins that mark the ILC2 lineage were expressed similarly in ILC2$_{10}$ and ILC2$_{act}$ (Fig. 3e, f). The data also indicated that ILC2$_{10}$ received an activation signal, as *Mki67*, *Klrg1*, *Il5*, *Il13*, and *Areg* were expressed similarly by ILC2$_{10}$ and ILC2$_{act}$ (Fig. 3e, g). *Maf*, a transcription factor that regulates IL-10 production in some cell types[26], was not differentially expressed by ILC2$_{10}$ (Fig. 3f). However, transcriptional regulators *Id3*, *Foxf1*, *Atf3*, and *Klf2* were more highly expressed by ILC2$_{10}$ (Fig. 3f). The differential expression of these genes was confirmed by RT-qPCR (Supplementary Fig. 2).

*Retnla*, encoding the secreted protein resistin-like molecule alpha (RELMα), which functions as a negative regulator of the T$_h$2 response during helminth infection[27] and is associated with alternatively activated macrophages[28], was more highly expressed in ILC2$_{10}$ (Fig. 3g, Supplementary Fig. 2). In contrast, *Tnf*, *Lta*, *Il2*, and *Ccl1* genes encoding pro-inflammatory effector molecules, were poorly expressed by ILC2$_{10}$ (Fig. 3g). Other differentially expressed genes that encode immunomodulatory

proteins included *Tnfsf11* (also expressed by ILC3[29]) and *Cxcl15*, the latter of which promotes neutrophil recruitment in the lung[30] (Fig. 3g). There are also likely signaling changes in $\text{ILC2}_{10}$, as

*Rgs18*, a negative regulator of G-protein-coupled receptor signaling implicated in platelet function and lineage choices in the bone marrow[31, 32], is downregulated in these cells.

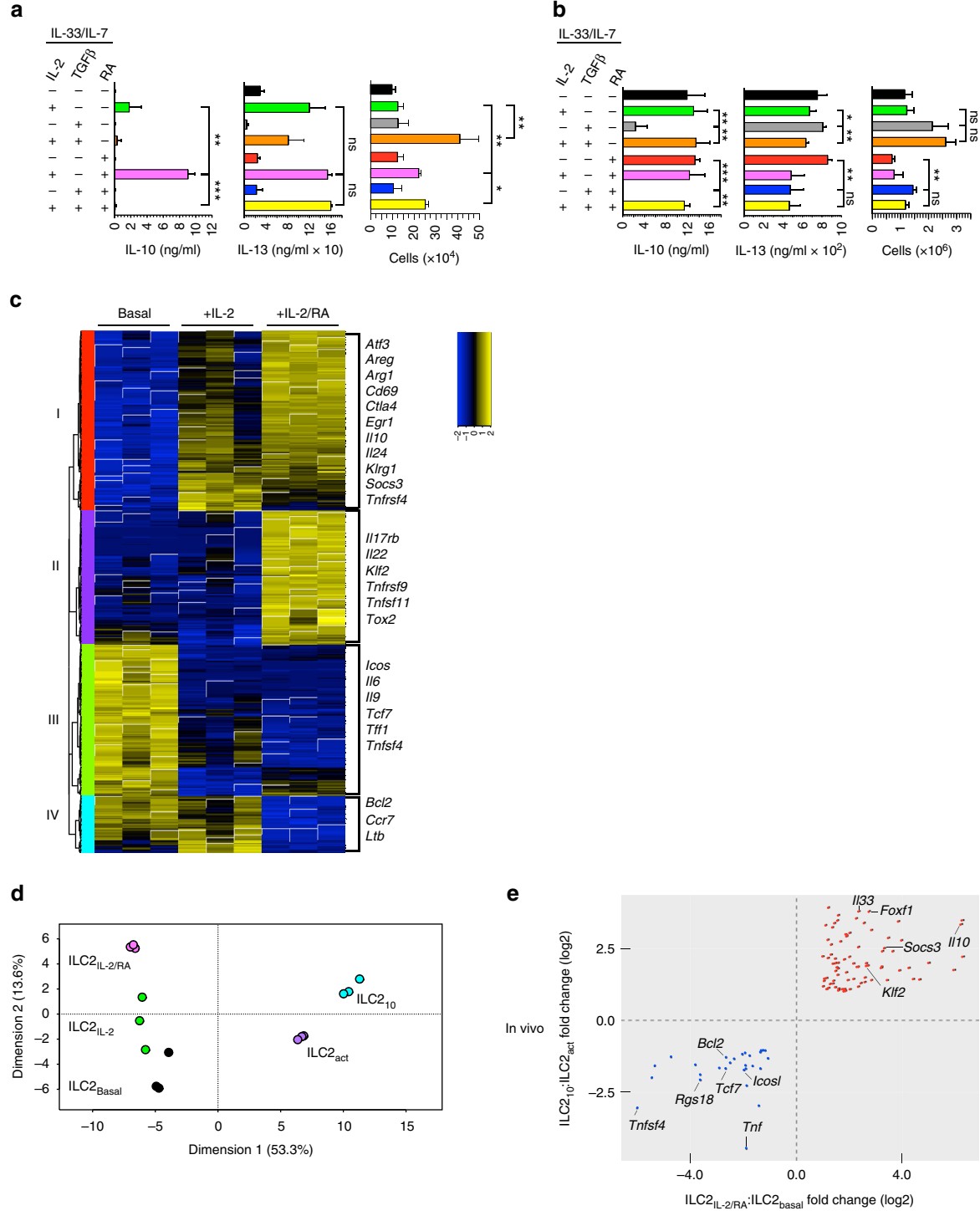

**Fig. 5** Expression profile of in vitro generated $\text{ILC2}_{10}$. **a**, **b** ILC2 from untreated (**a**) or IL-33-treated (**b**) wildtype mice were sorted and 5000 cells cultured for 6 days in the presence of IL-7 and IL-33 with the addition of indicated cytokine(s) and/or RA. Culture supernatants were harvested for IL-10 and IL-13 determination, and total cells were counted. **c** RNA-seq analysis of differentially expressed genes from in vitro stimulated ILC2 as indicated (ANOVA test with Benjamini and Hochberg correction). Examples of genes contained within each cluster (I, II, III, IV) are listed to the right of each cluster. **d** Principal component analysis of RNA-seq data presented in **c** and compared to isolated $\text{ILC2}_{10}$ or $\text{ILC2}_{act}$ from IL-33 injected mice. **e** Genes that were either upregulated (top right quadrant) or downregulated (bottom left quadrant) in indicated groups. Data are from three **a**–**e** independent experiments with two (**a**, **b**) or four (**c**–**e**) animals pooled per experiment. Error bars are standard deviation ***$P < 0.001$, **$P < 0.01$, *$P < 0.05$ and ns (not significant, $P \geq 0.05$) (Student's $t$ test)

IL-33 activated ILC2 upregulated expression of genes encoding the coinhibitory receptors CTLA4 and TIGIT (Fig. 1g). However, these genes were not differentially expressed by $ILC2_{10}$ and $ILC2_{act}$ (Fig. 3g), and thus may play a more general role in ILC2 homeostasis and/or function. Taken together, $ILC2_{10}$ down-regulate genes associated with inflammation and have a gene expression profile that is more similar to naïve ILC2, with the important exception of $Il10$.

**Contraction and recall of $ILC2_{10}$ in vivo.** To determine if $ILC2_{10}$ would be maintained in vivo in the absence of stimulus, we induced $ILC2_{10}$ formation in 10BiT$Foxp3^{GFP}$ mice with four daily injections of IL-33, and then analyzed at day 14 and day 30 (Fig. 4a). The expanded population of ILC2 persisted to day 30 (Fig. 4b, c). In contrast, there was near complete loss of cells with the $ILC2_{10}$ phenotype at both day 14 and day 30 (Fig. 4d, e), another distinguishing feature of this effector cell subpopulation. We considered the possibility that $ILC2_{10}$ effectors were lost from the lung due to migration to the draining lymph nodes (LN). However, while a small population of $ILC2_{10}$ was maintained in the lung at day 14 (Fig. 4d, Supplementary Fig. 3a), there was no substantial $ILC2_{10}$ population in the mediastinal LN at this time point (Supplementary Fig. 3a).

Alternatively, $ILC2_{10}$ might have extinguished $Il10$ expression and downregulated the reporter over the 2-week period. To address this, we re-challenged mice with a single injection of IL-33 at day 14 and analyzed the following day (Fig. 4a). In previously untreated mice, a single injection of IL-33 failed to induce production of $ILC2_{10}$ within 24 h (Fig. 4d, e compared to Fig. 2c, d) (PBS vs. single IL-33 injection; frequency of $ILC2_{10}$, $0.67 \pm 0.48\%$ vs. $0.46 \pm 0.48\%$; number of $ILC2_{10}$, $576 \pm 362$ vs. $313 \pm 287$). Total ILC2 did not significantly increase in frequency or numbers upon administration of recall IL-33 at day 14 (Fig. 4b, c). However, by day 30, the cells were sensitive to a single injection of IL-33, expanding an additional 1.7-fold on average (Fig. 4b, c). In contrast, at day 14, recall with a single injection of IL-33 revealed persistence of $ILC2_{10}$ effector cells, although well below starting levels (Fig. 4d, e). Like their day 4 counterparts, $ILC2_{10}$ at day 15 expressed more $Il10$ mRNA than reporter negative cells (Supplementary Fig. 3b). Similarly, a single injection of IL-33 at day 30 revealed maintenance of a small population of $ILC2_{10}$ that could be detected at day 31. Together, these data suggest contraction of the $ILC2_{10}$ population following removal of stimulus, with persistence of a subset of $ILC2_{10}$ that can be expanded and/or reactivated to produce IL-10 with minimal stimulation.

**In vitro generated $ILC2_{10}$ share similarities with those in vivo.** As only a subset of activated ILC2 differentiated into $ILC2_{10}$, we sought to determine the requirements for their generation. Ex vivo ILC2 cultured with IL-33 and the survival factor IL-7 expanded tenfold over 6 days and secreted IL-13, but failed to produce IL-10 (Fig. 5a). Others have shown that ILC2 activated in vitro with IL-33 and thymic stromal lymphopoietin also produce IL-13 and IL-5, but not IL-10[6]. Exogenous IL-2 has been shown to promote the in vivo expansion of lung ILC2, and can enhance IL-13 production[33]. The addition of IL-2 did not alter cell expansion in culture, but did lead to detectable IL-10 production and enhanced IL-13 secretion (Fig. 5a). As TGF-β and all-$trans$ retinoic acid (RA) modulate $T_{reg}$ cell differentiation from naïve CD4$^+$ T cells[34], we also tested the effects of these factors on cultured ILC2. No IL-10 was produced in cultures with addition of RA alone (Fig. 5a). However, addition of IL-2 and RA significantly increased IL-10 production with no effect on IL-13, and modestly increased cell yield when compared to stimulation of

cultures under basal conditions (Fig. 5a). TGF-β inhibited IL-2 and RA induced IL-10 expression, but also enhanced proliferation when added in combination with IL-2 (Fig. 5a). To determine how these factors would impact previously generated $ILC2_{10}$ effector cells, we sort purified ILC2 from IL-33-treated mice and cultured them under similar conditions (Fig. 5b). IL-13 and IL-10 were produced when activated ILC2 were cultured solely with IL-33 and IL-7, consistent with prior in vivo generation of $ILC2_{10}$. Addition of IL-2 or RA, or the combination of the two had little effect. TGF-β once again inhibited IL-10 production, although this effect was overcome by addition of IL-2.

To further dissect the molecular changes that occur during $ILC2_{10}$ generation, we cultured naïve ILC2 in vitro in the presence of $ILC2_{10}$ polarizing cytokines and performed RNA-seq analysis (Supplementary Data 3). Changes in gene expression could be grouped into four clusters; gene expression induced by IL-2 in the presence or absence of RA (cluster 1), genes upregulated specifically in the presence of RA (cluster 2), genes downregulated by IL-2 in the presence or absence of RA (cluster 3), and genes specifically downregulated in the presence of RA (cluster 4) (Fig. 5c). Genes encoding immunomodulatory proteins were found among all four clusters (Fig. 5c). As expected, $Il10$ was found in cluster 1, but $Il24$ and $Il22$ were contained in clusters 1 and 2, respectively, with the latter normally associated with ILC3. The cultured ILC2 also expressed $Gata3$ and $Rora$, but failed to express $Rorc$ or $Tbx21$, indicating that they remained ILC2 lineage cells (Supplementary Data 3).

Principal component analysis indicated that cultured ILC2 were genetically distinct from their in vivo counterparts (Fig. 5d), although there was some overlap between in vitro and in vivo generated $ILC2_{10}$ (Fig. 5e). Expression of the $Il9$ gene, which was not detected in $ILC2_{act}$ or $ILC2_{10}$ when generated in vivo (Fig. 3g), was expressed by cultured ILC2 but downregulated under $ILC2_{10}$ generating conditions (Fig. 5c). Both in vivo and in vitro generated $ILC2_{10}$ expressed $Il10$, but also had higher expression of transcriptional regulators $Foxf1$, $Klf2$, and $Atf3$ (Figs. 3f and 5c, e). In human macrophages, IL-10 acts in part by activating STAT3, resulting in downstream anti-inflammatory action[35]. SOSC3 is a key factor involved in the fine-tuning of pro- and anti-inflammatory responses by modulating STAT3 activity[36]. Both in vivo and in vitro $ILC2_{10}$ showed increased $Socs3$ expression (Figs. 3g and 5c, e). Other shared gene changes included downregulation of genes that are associated with inflammation such as $Tnfsf4$ and $Tnf$, as well as the anti-apoptotic factor $Bcl2$ (Fig. 5e).

**$ILC2_{10}$ generation is enhanced by IL-2.** IL-2 complexed with anti-IL-2-monoclonal antibody (IL-2c) expands cells that express the high affinity IL-2 receptor (CD25), including $T_{reg}$ cells[37]. As ILC2 express CD25 and addition of IL-2 to naive ILC2 induced $ILC2_{10}$ generation, we determined whether IL-2c would influence in vivo generation of $ILC2_{10}$ effector cells. Overall expansion of ILC2 was similar between mice injected with IL-33 in the absence or presence of IL-2c (Fig. 6a), yet the administration of IL-2c significantly increased the $ILC2_{10}$ population (Fig. 6b, c). Moreover, within the $ILC2_{10}$ population, $Il10$ reporter expression was significantly increased by IL-2c administration (Fig. 6d), although ST2 (IL-33R) expression was unchanged (Fig. 6e).

IL-33 induced both ILC2 expansion and upregulation of the $Il10$ gene in ILC2 from $Rag1^{-/-}$ mice (Fig. 6f, g), demonstrating that T cells were not required for $ILC2_{10}$ generation. There was no significant difference in $Il13$ gene expression in response to IL-33 in $Rag1^{-/-}$ mice, seemingly due to increased basal expression (Fig. 6h). Thus, the adaptive immune system may have an inhibitory role in the basal activation state of ILC2.

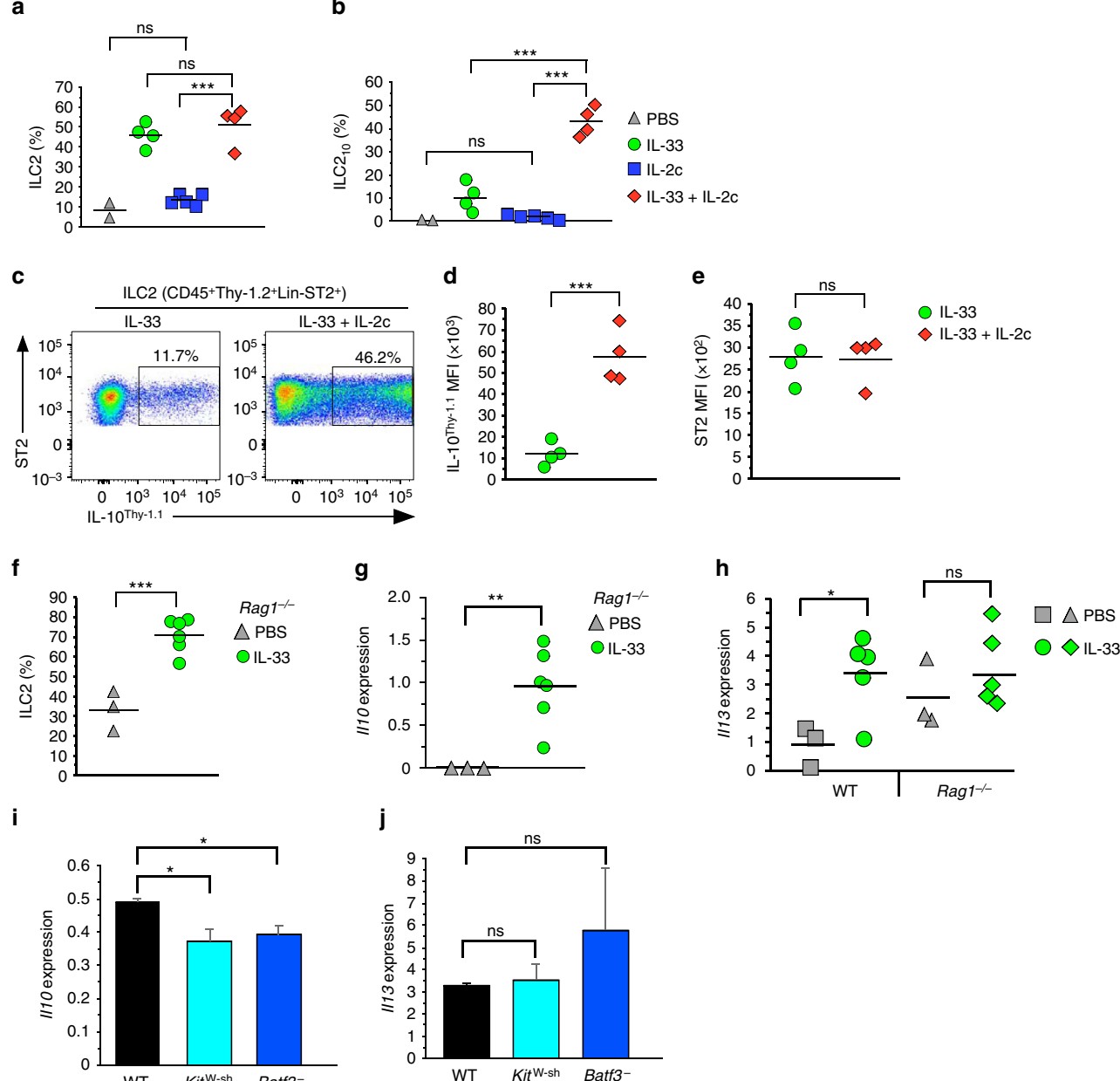

**Fig. 6** IL-2 complex enhances ILC2$_{10}$ effector cell generation in vivo. **a**, **b** Frequency of ILC2 (**a**) or ILC2$_{10}$ (**b**) from the lungs of 10BiTFoxp3$^{GFP}$ dual reporter mice treated for four consecutive days as indicated. **c** Flow cytometry analysis of ILC2$_{10}$ from mice injected with either IL-33 alone or IL-33 and IL-2c as a frequency of total ILC2 (Lin$^-$CD45$^+$Thy-1.2$^+$ST2$^+$). **d**, **e** Thy-1.1 (**d**) or ST2 (**e**) MFI, gated as in **c**. **f** Frequency of ILC2 within the CD45$^+$Thy-1.2$^+$ cell population from PBS or IL-33-treated Rag1$^{-/-}$ mice. **g**, **h** Quantitative RT-PCR analysis of Il10 (**g**) or Il13 (**h**) gene expression in ILC2 sorted from wildtype (WT) or Rag1$^{-/-}$ mice with indicated treatments. **i**, **j** ILC2 were sort purified from IL-33 injected animals deficient in either mast cells (Kit$^{W-sh}$) or CD103$^+$ dendritic cells (Batf3$^-$) and analyzed for Il10 (**i**) or Il13 (**j**) gene expression. Data are from two or three independent experiments with representative plot shown in **c**, compiled data on individual mice shown in **a**, **b**, **d**, **e**, **f**, **g**, **h**, and mean+standard deviation in **i**, **j**. ***$P < 0.001$, **$P < 0.01$, *$P < 0.05$ and ns (not significant, $P > 0.05$) (Student's t test)

Mast cells and CD103$^+$ dendritic cells (DC) have been described as in vivo sources of IL-2 and RA, respectively[38, 39]. To determine if ILC2$_{10}$ generation was compromised in the absence of these cell types, we injected mutant animals that lacked either mast cells (Kit$^{W-sh}$) or CD103$^+$CD11b$^-$ DC (Batf3$^-$) with IL-33 and analyzed the expression of the Il10 gene in isolated ILC2. In both mutant strains, there was a modest but significant decrease in Il10 expression from ILC2 (Fig. 6i) with no significant change in Il13 expression (Fig. 6j).

**In vivo generation of ILC2$_{10}$ reduces eosinophil recruitment**. ILC2-derived IL-5 and IL-13 play key roles in recruitment of eosinophils to lung, the latter cytokine acting via localized CCL11 (eotaxin 1) and CCL24 (eotaxin 2) production[40, 41]. To address the relationship between ILC2$_{10}$ generation and recruitment of eosinophils, we injected mice with IL-2c and IL-33 alone or in combination. As expected, IL-33 increased the number of lung eosinophils as compared to untreated animals (Supplementary Fig. 4a, b). Surprisingly, the administration of IL-2c in addition to

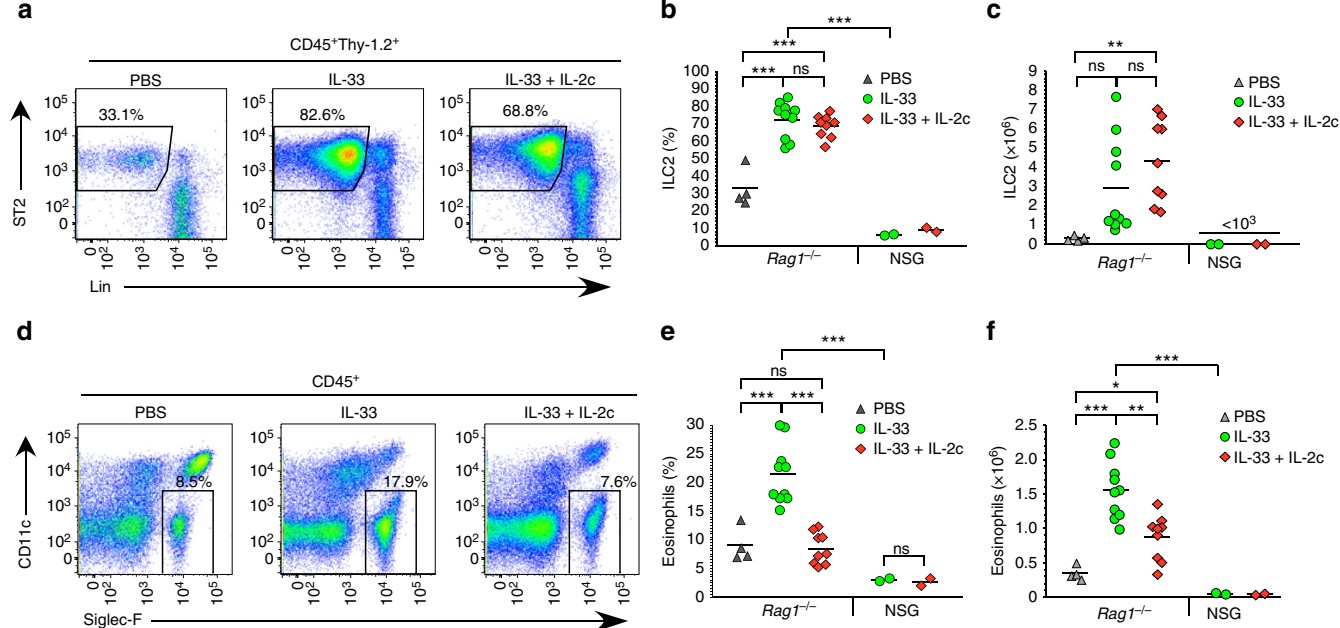

**Fig. 7** In vivo generation of ILC2$_{10}$ is associated with reduced eosinophil recruitment. **a** $Rag1^{-/-}$ mice were injected with the indicated cytokines for four consecutive days and analyzed for lung ILC2 the following day. **b**, **c** Compiled data of ILC2 frequency (**b**) and numbers (**c**) from $Rag1^{-/-}$ or NSG mice treated as in **a**. **d** $Rag1^{-/-}$ mice treated as in **a** analyzed for lung eosinophils. **e**, **f** Compiled data of eosinophil frequency (**e**) and numbers (**f**) from $Rag1^{-/-}$ or NSG mice treated as in **d**. Data are from one (NSG) or four ($Rag1^{-/-}$) independent experiments with representative plots shown in **a**, **d** and compiled data on individual mice shown in **b**, **c**, **e**, **f**. ***$P < 0.001$, **$P < 0.01$, *$P < 0.05$ and ns (not significant, $P > 0.05$) (Student's $t$ test)

IL-33 significantly reduced eosinophil accumulation (Supplementary Fig. 4a, b). This was not accompanied by any loss of IL-5 or IL-13 producing ILC2 in response to IL-2c, and indeed a greater proportion of ILC2$_{10}$ (favored by IL-2c treatment) were IL-5$^+$IL-13$^+$ double producers (Supplementary Fig. 4c). Others have reported that IL-2 and IL-33 increases CCL11 expression in lung parenchyma[41]. We also observed this, as well as a possible trend towards increased CCL24 that did not reach significance (Supplementary Fig. 4d, e). We also considered the possibility that despite low or absent expression of CD25 by eosinophils, that IL-2c might directly inhibit their migration. However, IL-2c had no effect on eosinophil migration to CCL11 in culture (Supplementary Fig. 4f).

To eliminate any confounding factors due to expansion of T$_{reg}$ cells or other T or B cells, we performed the same experiment with $Rag1^{-/-}$ mice, which lack adaptive immune cells but generate ILC2$_{10}$ (Fig. 6g). As a negative control, we used NSG mice which lack B cells, T cells, and ILC2, the latter due to due to lack of the common cytokine receptor gamma chain. IL-33 alone or in combination with IL-2c induced similar expansion of ILC2 in $Rag1^{-/-}$ mice (Fig. 7a–c). In contrast, while IL-33 increased lung eosinophils in $Rag1^{-/-}$ mice, animals treated with IL-33 and IL-2c had a significant reduction in lung eosinophilia (Fig. 7d–f). The frequency and number of eosinophils were extremely low in the lungs of treated NSG mice, pointing to the dominant role of ILC2 in driving eosinophil migration into lung tissue (Fig. 7d, e). Together, these data indicate that the extensive generation of ILC2$_{10}$ by IL-33 and IL-2c is associated with reduced eosinophil migration into the lung.

## Discussion

The parallels between CD4$^+$ T$_h$ effector cells and helper-like ILC have oft been noted, yet the generation of a specific IL-10 expressing effector population within the ILC2 lineage has not been reported. Here, we identified an ILC2 subpopulation, induced by IL-33 in vivo, that produces IL-10 and is molecularly distinct from activated IL-10$^-$ ILC2 and IL-10$^+$ T$_{reg}$ cells. In addition to high levels of the anti-inflammatory cytokine IL-10, we observed that ILC2$_{10}$ poorly express some pro-inflammatory genes when compared to ILC2$_{act}$. In vivo, these cells express *Retnla*, which is also anti-inflammatory in some model systems[5, 27, 28]. Moreover, there is no detectable expression of IL-9, as in some pro-inflammatory ILC2[42]. Therefore, ILC2$_{10}$ may prove to have anti-inflammatory properties. Recently, tumor infiltrating CD56$^+$CD3$^-$ ILC, possibly of NK cell origin, have been identified[43]. While these cells do not produce IL-10, they do possess the ability to suppress T cell cytokine production and expansion. In addition, ILC that secrete IL-10 and suppress ILC1 and ILC3 have recently been identified in the gut[44]. However, unlike ILC2$_{10}$, these regulatory cells express a distinct gene profile of transcription factors as compared to known ILC subsets, including lack of expression of *Gata3*. Together, the accumulating data points to great heterogeneity and likely functional complexity, including both pro- and anti-inflammatory activity, in the ILC compartment.

Interestingly, ILC2$_{10}$ were also produced in the lungs of mice upon chronic stimulation with the allergen papain. It is unknown what caused individual variability in the ILC2$_{10}$ response in this system. But the fact that the frequency of ILC2$_{10}$ strongly correlated with the extent of ILC activation and eosinophilia, suggests that ILC2$_{10}$ production is a byproduct of highly inflammatory conditions, as would be mimicked by direct injection of IL-33. This is consistent with an immunomodulatory role for ILC2$_{10}$.

ILC2$_{10}$ do not express *Foxp3* and are molecularly distinct from T$_{reg}$ cells, but it is unknown whether there is an alternative master regulator of this cell subset. Expression of the known regulators of ILC2 development and function are not differentially expressed in ILC2$_{10}$. However, we did observe higher expression of *Id3*, *Foxf1*, *Atf3*, and *Klf2* genes in ILC2$_{10}$ compared to ILC2$_{act}$. Id3 is

required for the TGFβ-induced generation of Foxp3[+] $T_{reg}$ cells[45]. All ILC2, including $ILC2_{10}$, highly express the related E protein inhibitor Id2, but the extent of functional overlap between Id2 and Id3 in this cellular context is not known. *Foxf1* is expressed in multiple mesenchyme-derived cell types and is critical for lung vascular development[46], but has no known role in the immune system. *Atf3* encodes a transcriptional repressor that dampens inflammatory responses, at least in part by inhibiting NF-κB activity[47]. KLF2 regulates NK cell (a cytolytic member of the ILC1 family) proliferation and survival[48]. Together, one or more of these factors may play a significant role in $ILC2_{10}$ identity.

It should also be noted that $ILC2_{10}$ are also unlikely to be molecularly closely related to IL-10 producing T-bet[+] Foxp3[−] CD4[+] Tr1 cells. Maf, AhR, and IRF4 are key transcription factors for IL-10 secretion and Tr1 cellular differentiation[49], and LAG-3 is a requisite marker[50]. By RNA-seq, $ILC2_{10}$ do not express LAG-3 or T-bet (encoded by *Tbx21*; Fig. 3f). Additionally, when comparing $ILC2_{10}$ and $ILC2_{act}$, *Maf* is not differentially expressed (Fig. 3f), expression of *Ahr* was 1.7-fold higher in $ILC2_{10}$ (and thus, did not reach our established threshold of differential expression), and *Irf4* was more highly expressed by $ILC2_{act}$ (Supplementary Data 2). This likely points to a distinct pathway for regulating IL-10 production and the effector cell program in $ILC2_{10}$.

One striking feature of $ILC2_{10}$ is their failure to fully down-regulate many genes associated with IL-33-activated ILC2, indicating an alternative activation pathway. In support of this, the signals to induce IL-10 from cultured ILC2 were distinct from those that elicited proliferation and IL-13 secretion. The latter required only IL-33 and IL-7, while IL-10 production was optimally induced by the addition of IL-2 and RA. Expression of *Retnla* is induced by IL-4 and IL-13[51], and although $ILC2_{10}$ produce both cytokines, they only express the IL-4 receptor. Thus, it is possible that an autocrine loop plays some role in the formation of $ILC2_{10}$ in vivo, although this would not preclude a significant role for other cell types.

In a model of allergic lung inflammation, IL-2 produced by adaptive immune cells induced ILC2 to make IL-9[42]. In contrast, *Il10* induction in ILC2 was independent of adaptive lymphocytes, as $ILC2_{10}$ were produced in $Rag^{−/−}$ mice. $ILC2_{10}$ generated in vivo also failed to express *Il9*, suggesting a distinct effector state and pathway of activation. Nevertheless, IL-2 presented in complex with anti-IL-2 monoclonal antibody, could act as a potent cofactor for $ILC2_{10}$ generation in vivo, increasing not only the proportion of $ILC2_{10}$, but also the expression of the *Il10* reporter on a per cell basis. The *Il2* gene itself was more highly expressed by $ILC2_{act}$ than $ILC2_{10}$. However, other cells in the lung besides T cells (and potentially ILC2) can produce IL-2, including IL-33 activated mast cells[38] and a population of CD2[+] ILC3-like cells[33]. Mice deficient in mast cells had a modest deficit in production of *Il10* by ILC2, supporting a role for mast cells, although whether that involves IL-2 secretion remains to be determined.

We also found that RA, which can be produced by CD103[+] lung dendritic cells[52], was a potent inducer of IL-10 in combination with IL-2 in vitro. Mice deficient in CD103[+] DC also showed some reduction in *Il10* gene expression in IL-33 activated ILC2. The data may indicate that a rather complex in vivo environment supports $ILC2_{10}$ generation, which can be mimicked at least in part by IL-2 and RA in vitro. RA regulates IL-22 production by γδ T cells and ILC3, and leads to attenuation of colitis[53]. We observed RA induction of *Il22* and *Il24* genes in ILC2 cultures, but did not detect *Il22* or *Il24* upregulation in vivo in response to IL-33 alone. In addition, TGF-β in combination with RA promotes $T_{reg}$ cell induction, yet inhibited IL-10 production by ILC2 without affecting IL-13 secretion. In total, these findings indicate complexity in ILC immune responses, and

suggest ways to manipulate ILC2 effector states in vitro and in vivo, which could prove useful in future therapeutic interventions.

The distinct gene expression pattern of $ILC2_{10}$ makes predictions of exact functions difficult, and may be very context dependent. Nevertheless, the gene expression pattern may indicate an anti-inflammatory role. In this regard, administration of IL-2c was highly effective at $ILC2_{10}$ generation and inhibited IL-33 mediated eosinophil recruitment to the lung. This was true even in the absence of an adaptive immune response. The underlying mechanism for this effect is unclear, as the reduction in eosinophilia was not correlated with a decrease in IL-5, CCL11, or CCL24, and eosinophil migration itself was not inhibited by IL-2c. IL-10 can inhibit allergen induced lung eosinophilia[54], although this may involve reduced IL-5 expression rather than a direct effect on eosinophils, which may poorly express the IL-10R[55]. Patients who receive IL-2 therapy can also exhibit eosinophilia, which evidence suggests is mediated by ILC2 production of IL-5[56]. Our data indicate greater complexity, and that there may be ways to achieve ILC2 activation without tissue recruitment of eosinophils.

The expanded population of IL-33 activated ILC2 had remarkable longevity over 30 days, even in the absence of continued stimulation. The bulk population of ILC2 were refractory to secondary stimulation at 14 days, but regained responsiveness at 30 days. In contrast, the $ILC2_{10}$ effector cell population underwent significant contraction as early as 14 days. The re-induction of $ILC2_{10}$ as a result of a single IL-33 injection as late as day 30, demonstrates that a subpopulation of these cells survived, but had switched off IL-10 production. There is a striking parallel in this pattern to antigen-driven T-cell responses that are characterized by expansion, contraction, and memory cell formation. Cytokine-induced memory-like NK cells have been described[57], and ILC2 have been shown to undergo contraction over long time periods, with enhanced responses upon secondary antigenic challenge in a model of asthma[58]. Given the results shown here, memory-like ILC2 may have important functions in not only the propagation of disease, but also the resolution of inflammation.

Variants of the *Il10* gene locus that cause decreased IL-10 production are positively associated with allergy, but are also negatively associated with helminth infection[59]. These processes also involve ILC2, and could suggest an anti-inflammatory role for ILC2-derived IL-10. The identification of a persistent ILC2 subtype that can produce IL-10 and other immunoregulatory proteins, and that can persist in vivo, may suggest a novel target to elicit the suppression of allergic and other pathogenic inflammatory immune responses.

## Methods

**Mice**. All mice were bred in house and kept under specific pathogen-free conditions. Mice used for experiments were between 6–14 weeks of age of either sex. *Il10*[21] and *Foxp3*[22] reporter strains were bred to produce 10BiT*Foxp3*[GFP] dual reporter mice. $Kit^{W-sh}$ (stock 012861), $Batf3^{−/−}$ (stock 013755), $Rag1^{−/−}$ (stock 002216), and NSG (stock 005557) mutant mice were obtained from the Jackson Laboratory. Mice were selected for experimental groups based solely on genotype within the age range indicated above. Within experiments, animals of each genotype were age-matched as closely as possible, within constraints of availability. All animal procedures were performed in accordance with protocols approved by the Cedars-Sinai Medical Center Institutional Animal Care and Use Committee.

**Antibodies and staining**. All samples were pre-incubated with anti-CD16/32 (39, 1:200; eBioscience) to block Fc receptors before staining. Lineage (Lin) antibodies directed against cell surface proteins used in the identification and isolation of all ILC2 populations are as follows (clone designations in parentheses, followed by dilution factor or concentration): CD8α (53-6.7, 1:2000), CD4 (GK1.5, 1:2000), CD3ε (145-2C11, 1:2000), γδTCR (eBioGL3, 1:1000), CD11b (M1/70, 1:1000), CD19 (1D3, 1:1000), B220 (RA3-6B2, 1:1000), Gr-1 (RB6-8C5, 1:1000), CD11c (N418, 1:500), NK1.1(PK136, 1:500), NKp46 (29A3.4, 1:500), Ter-119 (TER-119, 1:500), and FcεRII (B3B4, 1:500), all from eBioscience. Other antibodies used to

characterize ILC2 populations included Thy-1.2 (30-H12, 1:2000), ST2 (RMST2-33, 1:500), KLRG1 (2F1, 1:200), IL-5 (TRFK5, 0.25 μg per test), IL-13 (13A4, 0.25 μg per test), and ratIgG1κ isotype control (eBRG1, 0.25 μg per test) from eBioscience, and CD45 (30-F11, 1:500), and Thy-1.1 (OX-7, 1:2000) from BioLegend.

For intracellular cytokine staining, dissociated lung cells were incubated at 37 °C in EHAA medium (Irvine Scientific)/10% FCS containing 50 ng/ml PMA and 500 ng/ml ionomycin for 1 h. Brefeldin A (eBioscience) was then added to the wells at a final concentration of 3 μg/ml and incubated for 3 h. Cells were surface stained for CD45, Thy-1.2, ST2, Lin, and Thy-1.1 (Il10 reporter), followed by intracellular staining for IL-5 and IL-13, or isotype controls, using the Intracellular Fixation & Permeabilization Buffer Set (eBioscience).

For splenic eosinophil isolation, antibodies specified above and directed against B220, CD3, CD4, CD8α, and CD19, along with (BioLegend) CD115 (AFS98, 1:400) and CD49b (DX5, 1:400) were used for negative selection. Eosinophils were defined as CD45$^+$CD11c$^-$Siglec-F$^+$ (1RNM44N, 1:500), CD11b$^+$ cells.

Flow cytometry gating strategies are shown in Supplementary Fig. 5.

**Identification and isolation of lung cells.** Whole lung was removed following cardiac perfusion with PBS. Lungs were minced and digested in HBSS containing 1 mg/ml Liberase (0.1 mg/ml final). Cells were filtered using a 70 μm cell strainer and the red blood cells lysed. For IL-33-induced ILC2 activation in vivo, animals were injected intraperitoneally once daily with 500 ng IL-33 (BioLegend) or, as a control, with PBS for four days unless otherwise indicated. In some experiments, mice received daily intraperitoneal injections of IL-33 plus IL-2/anti-IL-2 complexes (IL-2c), made by mixing 1 μg recombinant murine IL-2 (eBioscience) and 5 μg anti-IL-2 monoclonal antibody (JES6-1A12) (Bio X Cell) and incubating for 30 min at 37 °C before co-injection with IL-33 in a final volume of 200 μl in PBS. Lung cells were stained for ILC2 or eosinophils with indicated antibodies, and analyzed using a LSRII or isolated using a FACS Aria III (BD Biosciences). Sort purities were >95%.

**RNA-seq and data analysis.** Lung ILC2 were isolated by cell sorting from IL-33 or PBS injected C57BL/6 or 10BiTFoxp3$^{GFP}$ mice. Cells were deposited into RNA-protect Cell Reagent (QIAGEN) and RNA was extracted using the RNeasy Plus Mini Kit (QIAGEN). In total, 10 ng of input RNA was used with the SMARTer Ultra Low Input RNA v3 kit (Clontech) to produce cDNA for downstream library preparation as previously described[15]. Briefly, the protocols for enzymatic fragmentation and ligation of the Ion Xpress Plus Fragment Library Kit (Life Technologies) were adjusted for our low-input total RNA of samples. Then resulting cDNA libraries were amplified onto Ion Sphere Particles with an Ion PI template OT2 200 Kit v3 (Life Technologies) and sequenced on an Ion Proton (Life Technologies) to generate an average depth of 20 million single-end 200 bp reads, with <6.7% of the reads from ribosomal RNA and over 94% of the reads mapping to the mouse genome. Similarly, libraries of ILC2$_{10}$ or ILC2$_{act}$ samples, or from IL-10$^+$ and IL-10$^-$ splenic T$_{reg}$ cells from IL-33 injected mice were constructed using SMARTer Ultra Low Input RNA v3 kit (Clontech), indexed using Nextera indices (Illumina) and sequenced on a NextSeq 500 (Illumina) using 75 bp single-end sequencing kit per the manufacturer's instructions. On average, about 10 million reads were generated from each sample. The raw reads were aligned by Tophat v2.1.0 to the mouse GRCm38 assembly with GENCODE M9 annotations downloaded from http://www.gencodegenes.org/. FPKM (fragments per kilobase of transcript per million mapped reads) values were calculated for lncRNA and protein coding genes with Cufflinks 2.2.1 software. FPKM levels below 1.0 were set to a "floor" value of 1.0. Principal component analysis (PCA) was used in an unsupervised gene expression analysis using FactoMineR v1.31 in R/Bioconductor v3.2. Two-tailed Student's t test was used to assess the significance of differences in gene expression between groups and then the Benjamini and Hochberg procedure was used for multiple test corrections[60]. A false discovery rate cutoff of 5% was used to select for significant differential expression genes. For in vitro ILC2$_{10}$ RNA-seq, lung ILC2 were isolated as described and cultured in vitro for 6 days with select cytokine(s) alone or in combination. FPKM values for the three replicates were averaged in each group. ANOVA test was performed for the genes if the average FPKM for any one of the groups was at least four-fold higher than one of the other two groups. P values calculated from ANOVA test for each gene and were corrected by calculation of the q-value by the Benjamini and Hochberg method. Genes with a significant difference in expression among the three groups (FDR < 1%) were used for the heat map generated in Fig. 5.

**Cell cultures.** ILC2 were isolated by sorting cells directly into complete DMEM containing IL-7 (20 ng/ml) and IL-33 (20 ng/ml) (Biolegend). In total, 5,000 cells per well per condition were plated in a 96-well plate and supplemented with combinations of human IL-2 (10 ng/ml) (Peprotech), all-trans retinoic acid (RA) (1 μM) (Sigma-Aldrich) and TGF-β1 (10 ng/ml) (BioLegend). Cells were fed on day 3 by adding an equal volume of fresh media supplemented with appropriate cytokines and/or RA. At day 6, total cell counts were performed and the culture supernatants were analyzed for IL-10 and IL-13 by ELISA (eBioscience).

**Allergic response to papain.** Mice were anesthetized by isoflurane inhalation, followed by the intranasal administration of 30 μg of papain in a total of 12 μl to

each nostril. Papain was administered once a day for five days (d0–d4), followed by discontinuation of treatment (d5–d13). Papain was then re-administered for another 5 days (d14–d18), followed by analysis of lungs 3 days later (d21).

**Eosinophil migration.** Eosinophils were enriched from mouse spleen by negative selection by staining with biotin-conjugated antibodies as above and using the EasySep Mouse Streptavidin RapidSpheres Isolation Kit (STEMCELL Technologies). Enriched cell populations were then stained and sort purified (CD45$^+$SiglecF$^+$CD11c$^-$CD11b$^+$). 1–1.5 × 10$^4$ purified eosinophils in 80 μl pre-warmed RPMI-1640 media containing 5% bovine serum albumin were placed in the upper compartment of wells containing HTS 96-well Transwell Permeable Supports (Corning). The lower compartment contained various concentrations of CCL11 (BioLegend) in the absence or presence of IL-2c (0.2 μg IL-2 equivalent) in 235 μl. Cells were allowed to migrate at 37 °C in a humidified 5% CO$_2$ chamber for 2 h, the plate placed on ice for 10 min, and then centrifuged at 200 × g for 10 min to detach cells from the membranes. The transwells were then removed and the media containing cells in the bottom compartment was collected. The supernatant was centrifuged for 5 min at 200 × g, decanted, and the cell pellets were resuspended in 20 μl of AO/PI staining solution and counted on the Cellometer Auto 2000 cell viability counter (Nexcelom). Data are expressed as the average percentage of input cells that migrated to the bottom chamber, from duplicate wells.

**Lung chemokine assays.** Left lungs were digested in 1 ml of PBS in a gentleMACS Octo Dissociator (Milltenyi Biotec) with heaters, running program m_LDK_1. Cells and cellular debris were removed by passage through a 70 μm cell strainer (BD Biosciences) and centrifugation at 200 × g for 5 min at 4 °C. Concentrations of CCL11 and CCL24 in the supernatant were determined by ELISA (Abcam or BioLegend).

**qRT-PCR.** cDNA was generated using Superscript VILO (Life Technologies) and PCR performed using QuantiTect SYBR green (QIAGEN). All primer sets were purchased from QIAGEN (QuantiTect), and gene expression was normalized to Hprt or Gapdh within each experiment.

**Statistics.** Means, standard deviations, and the probability (P) associated with a Student's t test using a two-tailed distribution of equal variance are shown in some figures. P values of <0.05 were considered to represent means with a statistically significant difference. Statistical analysis was performed on groups with similar variance, and limited variance was observed within sample groups. Sample or experiment sizes were determined empirically for sufficient statistical power. No samples were excluded specifically from analysis.

**Data availability.** Sequence data that support the findings of this study have been deposited in Gene Expression Omnibus at NCBI with the primary accession code GSE81882 (https://www.ncbi.nlm.nih.gov/geo/query/acc.cgi?acc=GSE81882).

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

## Acknowledgements

The authors thank the Cedars-Sinai Medical Center Flow Cytometry Core, the Cedars-Sinai Genomics Core, and the Cedars-Sinai Medical Center Biomedical Sciences and Translational Medicine Graduate Program. We also thank C. Weaver (University of Alabama, Birmingham), V. Kuchroo (Harvard Medical School) and G. Martins (Cedars-Sinai Medical Center) for the kind gifts of mice. This work was supported by the U.S. National Institutes of Health (Grants 5R21AI124209 and 2R01AI054977 to J.K.).

## Author contributions

C.R.S., A.K., J.K.: conceived and designed the experiments, and analyzed the data. C.R.S., A.K., B.d.l.T., A.R.Y.: performed the experiments. Y.W., J.T.: oversaw RNA-seq and analyzed the resulting data. C.R.S., J.K.: wrote the manuscript.

## Additional information

**Competing interests:** The authors declare no competing financial interests

