## [Peer Review File · Nature Communications]

Reviewers' comments:

Reviewer #1 (Remarks to the Author):

The manuscript entitled "An alternative activation pathway drives the generation of IL-10-producing innate lymphoid type-2 effector cells", investigated the changes in lung ILC2 gene expression by RNAseq revealing the induction of IL-10 expression by these cells upon challenge with IL-33 in vivo. Using 10-BiT IL-10 reporter mice, the authors revealed that IL-10 positive and negative ILC2 subpopulations have a distinct gene expression signature. Induction of IL-10 expression by IL-33 activated ILC2 is augmented by IL-2 but appears to be independent of the adaptive immune response in vivo. In vitro, the production of IL-10 by untreated ILC2 cells cultured with IL-33 requires IL-2 and is further enhanced by retinoic acid but inhibited by TGF- β . TGF- β also inhibits the production of IL-10 by IL-33 activated ILC2 which can be overcome by IL-2. This study further shows that increased IL-10 positive ILC2 may correlate with decreased accumulation of eosinophils in the lungs after treatment with IL-33 plus IL-2, although the evidence for IL-10 positive ILC2 ultimately inhibiting eosinophil accumulation is lacking.

While this study is of interest, it is predominantly of a descriptive nature and the data as yet do not support the conclusions of the paper.

Major Comments:

1) Identification of ILC2 - it is not clear how ILC2 were identified and sorted. The authors must provide a list of all the markers used to identify ILC2 (including all "Lin" markers). The numbers of ILC2 in the lungs of mice treated with PBS reported in Fig. 1c seem 10 times higher than the numbers of ILC2 in the lungs of naïve mice (C57BL/6 WT or Rag1^{-/-} mice, and Balb/c mice) reported in previous studies (Monticelli et al., Nat Immunol 2011; Chang et al., Nat Immunol 2011). The authors should clarify this difference in ILC2 numbers with respect to those described in the literature to ensure the specificity of the cells they are isolating.

2) The authors must include information in each figure legend regarding the number of times the experiments were performed and the size of the groups to ensure statistically significant and reproducible findings.

3) In Fig. 3, it would be more logical to show the differentially expressed genes as Fig. 3c (currently Fig. 3g), followed by the selected genes (currently Fig. 3c-e). In Fig. 3f, it is not clear if the data were obtained from same experiments of RNAseq or independent experiments and why only 3 values per group are shown for Id3 while all other genes have 4 values per group? Also in Fig. 3f, it is not clear why only these genes were validated by RT-PCR? Similarly expressed genes (such as Il5 and Il13, line 91) and genes poorly expressed by ILC2-10 (such as Tnf and Il2, line 104) should be validated.

4) In Fig. 4, data for different time-points should be presented in a chronological order (i.e., d1 (green) - d4 (black) - d14 (blue) - ... ; instead of d4 (black) - d1 (green) - d14 (blue) -). Control group - d0 without treatment - is missing in order to determine if indeed a single injection of IL-33 fail to induce significant expansion of total or IL-10-expressing ILC2 (as stated by the authors in lines 121-122).

5) Also in Fig. 4, it is not clear which mice were used in the experiment and whether as throughout 10BiT mice were used. This is very important since 10BiT-transgenic mice stably identifies cells in which IL-10 has been previously activated and those cells actively transcribing Il10 (Maynard et al., Nature Immunology), and so once induced the IL-10 as assessed by the 10-BiT reporter will stay on. Thus since they see a decrease in the IL-10-expressing cells (presumably using the 10-BiT reporter mice) what is actually happening to Thy1.1⁺ (ie IL-10⁺) ILC2 between d4 and d14, since the expression of Thy1.1 is stable and also present in cells in which IL-10 has been previously activated. Are these cells more prone to die and/or migrate to other organs compared to Th1.1neg

ILC2 and does IL-33 merely recruits such IL-10-expressing ILC2? Also the frequency and numbers of Thy1.1+ ILC2 may increase after 1 injection of IL-33 from d14 to d15 (and from d30 to d31) because cells in which IL-10 has been previously activated (Thy1.1+) proliferate and does not necessarily mean that these cells are expressing IL-10 at that specific time in response to IL-33 treatment. IL-10 mRNA levels in sorted ILC2¹⁰ cells should be quantified at the different time points to clarify this.

6) Fig. 5 a and b, it is stated in the materials and methods and figure legend that 5,000 cells were plated per well and cultured for 6 days. At the end of the culture, the cells were counted and numbers noted in the third panels of the figure. These numbers suggest a very fast division rate for untreated ILC2 cells (Fig. 5a), which was further increased in the IL-33 activated ILC2 cells (Fig. 5b). The authors should comment on this observation.

7) For Fig. 7b, it would be very informative if the total cell numbers were also shown. IL-2c treatment may be inducing expansion of T cells that may result in decreased frequency of other cell types and so masking the accumulation of eosinophils. What happens to the expression of IL-5 and IL-13 upon IL-33 treatment in the presence of IL-2c?

8) In Fig. 7, the authors should confirm the increased expression of IL-10 by ILC2 after treatment with IL-33 plus IL-2c by mRNA level (as in Fig. 6h). The findings in Fig. 6 and 7 would be greatly complemented by IL-2 blocking experiments, which, if the hypothesis is correct, should block the production of IL-10 by ILC2 cells upon IL-33 treatment and further increase the observed eosinophilia – the opposite to the IL-2c treatment. The role of IL-10 in reducing eosinophil recruitment to the lung could be addressed using Rag-/-IL-10-/- (double ko) mice or using IL-10 or IL-10R neutralizing Abs.

9) Although of interest it would good for the authors to demonstrate that ILC2 can be induced to make IL-10 in a more physiological setting such as allergy. In addition, it would be helpful to compare then the production of IL-10 by ILC2 to that produced by classical TH2 cells in this setting.

Minor points:

- 1) The authors must clarify which mice were used to obtain data from Fig. 1 and the background of all mutant (reporter and ko) mice used in the study.
- 2) Purity of sorted ILC2 should be reported.
- 3) Gate in Fig. 2b (left) requires adjustment (and subsequently Fig. 2c and d) as it doesn't include all Foxp3+ IL-10+ cells.
- 4) Primer sets used for qRT-PCR should be included as well as the calculation method.
- 5) Sup. Table 1 is inaccurately indicated in line 97
- 6) Fig. 5c, larger font should be used for the key. Also, the representative genes shown at the edge of the heatmap are somewhat misleading.
- 7) Fig. 5d is inaccurately indicated in line 168
- 8) Lines 196-197: it is not clear what the authors mean by "However, we found that these signals can also induce ILC2-10"?

Reviewer #2 (Remarks to the Author):

This manuscript describes a previously unappreciated IL-10 producing ILC2 generation. The scientific is sound and well designed but some modifications in the text or in the figures would be appreciated as detailed below.

Abstract:

Page 2 lines 1-2: "Type-2 innate lymphoid (ILC2) cells share cytokine production and transcription

factor expression with CD4+ Th2 cells.”

Page 2 line 2: “but it is unknown if ILC2 cells can undergo additional functional diversification.”

This sentence is confusing: what about plasticity like the inflammatory ILC2 that are able to produce IL-17 when cultured with IL-2 and IL-7 (Huang et al, Nat Immunol 2015) for example?

Page 2 lines 11-12: “These data demonstrate previously unappreciated heterogeneity in ILC2 cell responses.” In line with the previous remark, some heterogeneity in ILC2 responses has already been demonstrated such as IFN- γ production by ILC2 during lung inflammation. A relative play down such as “a previously unappreciated IL-10 producing ILC2 generation” would be appreciated.

Introduction:

Page 3 lines 15-16: “ILCs, have been identified in mice and humans and have many parallels to CD4+ helper T (Th) effector cell subsets.” It would be more accurate to speak of helper-like ILC rather than ILC OR of T cells in general rather than CD4+ Th because ILC include the cytotoxic NK.

Page 3 lines 16-18: “In this regard, Type-1 ILCs (ILC1 cells), Type-2 ILCs (ILC2 cells), and Type-3 ILCs (ILC3 cells) have been compared to Th1, Th2, and Th17 cells, respectively.” The actual nomenclature includes NK cells in the Type-1 ILCs group and LTi in the Type-3 ILCs group.

Page 3 lines 20-23: Although this difference in cell generation is important, it should not be undermined that other differences are important as well such as the lack of antigen-specific receptors derived from gene rearrangements dependent on RAG recombinases or the tissue-residency of ILC that are rare in the circulating blood and represent a minority of lymphoid tissue cells.

Page 3 lines 24-25: “ILC2 cells have a beneficial role in eradication of parasitic helminthes.” A reference is missing there.

Results:

Page 4 lines 52-54: “Genes encoding cell surface molecules used for cell isolation (Ptprc, Thy1, and Il1rl1) or markers of the ILC2 cell type (H2-Ab1, Icos, Il2ra) were expressed similarly in both cell populations.” What about klr1 and il17rb both surface markers also expressed by ILC2 that are labeled in red in the figure 1e?

Page 6 lines 86-88: What about the difference between IL10+ Th1 and IL10- Th1?

Discussion:

Page 11 lines 209-210: The parallels between CD4+ Th effector cells and helper-like ILCs have oft been noted, yet identification of diversified effector populations within the ILC2 lineage has not been reported.

Figures:

Figure 1b and c: The color code should be changed because blue and red both reminds of all the graphs were differentially expressed genes are labeled in red.

Figure 3a: Normally, the contribution of each variable to all the dimensions of the PCA can be determined. This could be interesting to look at in order to understand what are the variables contributing the most to the 2nd dimension, meaning, what are the variables upon which there is the difference between ILC233, ILC2act and ILC210 that are not differentiated by the 1st dimension.

Figure 3f: In line with the previous remark, the color code is confusing because the ILC2act are in blue although they were represented as green dots in the principal component analysis graphs Figure 3a and b.

Figure 4b,c,d and e: It could be informative to look at the amount of IL-10 produced by ILC2, ILC210 with the same design of experiment to see if the cells are more potent of IL-10 production when recalled 14 or 30 days after the first induction.

Figure 5d : Same remark as previously, it would be interesting to know what are the variables contributing the most to the 1st and 2nd dimension of the PCA, meaning, the variable differentiating the populations represented.

Methods:

Antibodies and flow cytometry: somewhere should be precised what antibodies the lineage cocktail includes.

Response to Referees

This paper presents the discovery of a new subset of IL-10 producing innate lymphoid cells with a unique gene expression signature and the potential to possess immunoregulatory properties *in vivo*. Although plasticity within some of the ILC lineage has been described, mostly involving upregulation of T-bet and conversion to an ILC1-like cell, specific effector subtypes within the ILC2 cell lineage have not been identified. We think these findings are therefore of fundamental importance and reveal an 'adaptive' quality to ILC2 cell responses, with potential future therapeutic import.

We thank the reviewers for their critiques that have enabled us to add a significant amount of new data, including induction of ILC2₁₀ cells by allergen, resulting in greater depth and impact to the study. The specific issues raised by the reviewers are addressed below, and significant changes to the manuscript are highlighted.

Reviewer 1

“It is not clear how ILC2 were identified and sorted. The authors must provide a list of all the markers used to identify ILC2 (including all “Lin” markers).”

We thank the reviewer for pointing out this oversight. ILC2 were isolated using the following lineage markers: CD8 α , CD4, CD3, $\gamma\delta$ TCR, CD11b, CD19, B220, Gr-1, CD11c, NK1.1, NKp46, Ter-119, and Fc ϵ R2. This information was added to the methods section with antibody clones included. In addition, RNA-seq analysis indicates that these cells are of the ILC2 cell lineage as assessed by expression of genes encoding cell surface proteins (**Fig. 3e**), transcriptional regulators (**Fig. 3f**) and other associated activation markers (**Fig. 3g**), while negative for those associated with cell types of other lineages. The exact method for analysis and isolation of ILC2 can be found in¹.

“The numbers of ILC2 in the lungs of mice treated with PBS reported in Fig. 1c seem 10 times higher than the numbers of ILC2 in the lungs of naïve mice (C57BL/6 WT or Rag1^{-/-} mice, and Balb/c mice) reported in previous studies (Monticelli et al., Nat Immunol 2011; Chang et al., Nat Immunol 2011). The authors should clarify this difference in ILC2 numbers with respect to those described in the literature to ensure the specificity of the cells they are isolating.”

Counts of ILC2 cells are higher due to increased isolation efficacy through refined isolation techniques and improvements to early published protocols. These numbers are in line with recent publications (see specifically, Fig. 3k in ref. 1 and Fig. 2b in ref. 2).

“The authors must include information in each figure legend regarding the number of times the experiments were performed and the size of the groups to ensure statistically significant and reproducible findings.”

We apologize for this oversight. Specific information regarding the number of experiments and number of animals used is included at the end of each figure legend. The vast amount of data is shown for individual mice or pooled samples, rather than simply averaged.

“In Fig. 3, it would be more logical to show the differentially expressed genes as Fig.3c (currently Fig. 3g), followed by the selected genes (currently Fig. 3c-e).”

The order of the figure parts was changed as recommended and is reflected appropriately in the figure legend and results section.

“In Fig. 3f, it is not clear if the data were obtained from same experiments of RNAseq or independent experiments”.

The data shown in Fig. 3f was generated from independent experiments. These data have now been moved to supplementary data with a separate figure legend that clarifies this point (**See Supplementary Fig. 2**)

“and why only 3 values per group are shown for Id3 while all other genes have 4 values per group?”

Unfortunately, due to the scarcity of ILC2₁₀ cells compounded further by yields from stringent cell sorting, *Id3* qRT-PCR was quantitated using only 3 mice due to lack of sufficient material to test for all 4 genes. We do not believe this has any impact on the conclusions of the manuscript.

“Also in Fig. 3f, it is not clear why only these genes were validated by RT-PCR? Similarly expressed genes (such as Il5 and Il13, line 91) and genes poorly expressed by ILC2-10 (such as Tnf and Il2, line 104) should be validated.”

We solely focused on transcriptional regulators that might impact ILC2₁₀ development and one biological mediator that we thought would be of interest to the community based on previously published data. There are many changes in gene expression and we cannot validate them all, nor do we think that is necessary with highly quantitative RNA-seq. Future follow-up on any of these genes would certainly require this data, but here we are most interested in the uniqueness of the gene signature rather than the function of any particular gene in this context.

“In Fig. 4, data for different time-points should be presented in a chronological order (i.e., d1 (green) – d4 (black) – d14 (blue) - ... ; instead of d4 (black) – d1 (green) – d14 (blue) -).”

We have revised this figure as requested for clarity.

“Control group - d0 without treatment - is missing in order to determine if indeed a single injection of IL-33 fail to induce significant expansion of total or IL-10-expressing ILC2 (as stated by the authors in lines 121-122).”

Unstimulated animals were assessed in **Fig. 2c,d** and determined to have no difference between animals that were injected one time with IL-33 as presented in **Fig. 4**. The frequencies and numbers as well as corresponding standard deviations are as follows:

PBS injected mice, ILC2₁₀:

Frequency: 0.67% (+/-0.48%)

Number: 576 (+/-362)

Single IL-33 injection, ILC2₁₀:

Frequency: 0.46%(+/-0.48%)

Number: 313 (+/-287)

Students t-test:

Frequency: 0.59

Number: 0.49

“Also in Fig. 4, it is not clear which mice were used in the experiment and whether as throughout 10BiT mice were used. This is very important since 10BiT-transgenic mice stably identifies cells in which IL-10 has been previously activated and those cells actively transcribing Il10 (Maynard et al., Nature Immunology), and so once induced the IL-10 as assessed by the 10-BiT reporter will stay on.”

“Thus since they see a decrease in the IL-10-expressing cells (presumably using the 10-BiT reporter mice) what is actually happening to Thy1.1+ (ie IL-10+) ILC2 between d4 and d14, since the expression of Thy1.1 is stable and also present in cells in which IL-10 has been previously activated. Are these cells more prone to die and/or migrate to other organs compared to Th1.1neg ILC2 and does IL-33 merely recruits such IL-10-expressing ILC2?”

In this figure, 10BiT reporter mice were used. This has been clarified in the text and figure legend. We respectfully disagree that it is a given that the reporter stays on for 2 weeks or a month. With removal of stimulus these cells may no longer be transcribing the reporter and presumably the mRNA has decayed in that time period (the average half-life of mammalian mRNA is ~9 hours³). We are not aware of studies that have measured the half-life or turnover of cell surface Thy-1 protein in this context for these extended time periods. We have suggested that there is contraction of this population (meaning cell death, similar to contraction during a T cell immune response), which also fits with reduced Bcl2 in these cells (**Fig. 3g**) and the fact that even with restimulation by IL-33, the ILC2₁₀ cell numbers remain less than those seen at day 4. This was the major reason we performed reactivation and looked at day 15, i.e. are the cells still there but simply not expressing the reporter? The data indicate that there is an existing but reduced population of cells that can express the reporter at these late time points. And importantly, the ILC2₁₀ cells behave very differently from the bulk ILC2 cell population upon removal of stimulus.

However, we agree with the reviewer that there could be changes in migratory properties of the cells. This is difficult to disprove without very extensive adoptive transfer experiments involving very small cell populations, but we did ask if the loss of cells at day 14 might be associated with an enrichment of the cells in draining LN. This was not the case (**Supplementary Fig. 3a**), and thus we continue to favor a T cell-like contraction.

“Also the frequency and numbers of Thy1.1+ ILC2 may increase after 1 injection of IL-33 from d14 to d15 (and from d30 to d31) because cells in which IL-10 has been previously activated (Thy1.1+) proliferate and does not necessarily mean that these cells are expressing IL-10 at that specific time in response to IL-33 treatment. IL-10 mRNA levels in sorted ILC210 cells should be quantified at the different time points to clarify this.”

To address this, we sort purified IL-33 re-challenged reporter positive and reporter negative ILC2 cells at day 15. Similar to day 4, *I10* mRNA was enriched in the reporter positive cells at day 15 (**Supplementary Fig. 3b**). How much of the increase in ILC2₁₀ cells within a day is cell proliferation compared to reexpression of the reporter is not clear. But it is worth noting that a single dose of IL-33 does not expand ILC2 cells within this same time period (see above), so regardless, our data demonstrate that a small population of ILC2₁₀ cells are maintained and can respond to IL-33 either proliferatively and/or by reactivation of the *I10* gene.

“Fig. 5 a and b, it is stated in the materials and methods and figure legend that 5,000 cells were plated per well and cultured for 6 days. At the end of the culture, the cells were counted and numbers noted in the third panels of the figure. These numbers suggest a very fast division rate for untreated ILC2 cells (Fig. 5a), which was further increased in the IL-33 activated ILC2 cells (Fig. 5b). The authors should comment on this observation.”

The fast division rates of ILC2 are consistent with doubling times presented in *Moro et al.*⁴ (their **Fig. 1f**). Note that there are no ‘untreated’ cells in this experiment. All cells were cultured under basal stimulatory conditions (IL-33 and IL-7) to ensure proliferation and survival. As these conditions did not lead to IL-10 production, we then tested the effects of addition of other cytokines or retinoic acid (RA).

“For Fig. 7b, it would be very informative if the total cell numbers were also shown. IL-2c treatment may be inducing expansion of T cells that may result in decreased frequency of other cell types and so masking the accumulation of eosinophils.”

We have focused on RAG1^{-/-} mice to address this, as these animals eliminate any confounding variables from stimulation of the adaptive immune system. Numbers of ILC2 and eosinophils are now included in **Fig.**

7. Along with analysis of greatly expanded numbers of animals, these data strengthen the conclusions. In addition, we have added NSG mice as control (**Fig. 7**), which demonstrate that the increase of eosinophils in response to IL-33 is ILC2 cell dependent.

With the focus on RAG1^{-/-} mice we have moved our original observation on wildtype mice to **Supplementary Fig. 4** (including greater numbers of animals). **Fig. S4** also presents measurements of eotaxin 1 and 2 (CCL11 and CCL24, respectively) levels in the lung in response to IL-33 +/- IL-2c, and the lack of a direct effect of IL-2c on eosinophil migration in culture.

“What happens to the expression of IL-5 and IL-13 upon IL-33 treatment in the presence of IL-2c?”

We addressed this by intracellular cytokine staining, presented in **Supplementary Fig. 4c**. There are similar patterns of staining with *in vivo* administration of IL-33 in the presence or absence of IL-2c. However, there was a greater frequency of IL-13/IL-5 double producers among ILC2₁₀ cells. This, along with eotaxin 1 and 2 levels, does not easily explain the reduction of eosinophils.

“The findings in Fig. 6 and 7 would be greatly complemented by IL-2 blocking experiments, which, if the hypothesis is correct, should block the production of IL-10 by ILC2 cells upon IL-33 treatment and further increase the observed eosinophilia – the opposite to the IL-2c treatment. The role of IL-10 in reducing eosinophil recruitment to the lung could be addressed using Rag^{-/-}IL-10^{-/-} (double ko) mice or using IL-10 or IL-10R neutralizing Abs.”

We agree that these are interesting experiments, but feel that these are outside the scope of this paper, given the extensive other data we have presented to elucidate the discovery of these cells.

“Although of interest it would good for the authors to demonstrate that ILC2 can be induced to make IL-10 in a more physiological setting such as allergy.

We thank the reviewer for this suggestion and agree that the production of ILC2₁₀ cells in a more physiological setting would enhance the impact of the study. As such, we tested the production of ILC2₁₀ cells during acute and chronic responses to the allergen papain. Notably, we failed to find ILC2₁₀ cells upon our various attempts with acute treatment with papain (not shown), but did under chronic stimulatory conditions, which also led to eosinophilia as expected (**Fig. 2g,h, Supplementary Fig. 1a,b**). Most interestingly, we found variability among individual mice for production of ILC2₁₀ cells, which strongly correlated with the extent of eosinophilia and the activation state of the overall population of ILC2 cells (**Supplementary Fig. 1c,d**). We interpret this finding as indicating that ILC2₁₀ generation is favored by highly inflammatory conditions, which is consistent with a potential immunomodulatory role for these unique effector cells.

Minor points

“The authors must clarify which mice were used to obtain data from Fig. 1 and the background of all mutant (reporter and ko) mice used in the study.”

Figure 1 data is from wildtype (C57BL/6) mice treated with IL-33 or PBS vehicle. This information has been added to the figure legend.

“Purity of sorted ILC2 should be reported.”

Post-sorting tests indicated purity of >95%. This has been added to methods.

“Gate in Fig. 2b (left) requires adjustment (and subsequently Fig. 2c,d) as it doesn’t include all Foxp3+ IL-10+ cells.”

While we can move the gates and recalculate the data to include potentially more ILC2₁₀ cells, we prefer the more conservative gate. Moving the gate to the left, while picking up very low expressing cells (of unknown biological significance), also begins to impinge on the major negative population (activated cells may also have somewhat higher autofluorescence, hence the modest shift to the right). Thus, we think this is likely to, if anything, make the data less robust and less biologically relevant.

“Primer sets used for qRT-PCR should be included as well as the calculation method.”

cDNA was generated using Superscript VILO (Life Technologies) and PCR performed using QuantiTect SYBR green. All primer sets were purchased from QIAGEN (QuantiTect), and gene expression was normalized to a housekeeping gene. This information was added to the methods.

“Sup. Table 1 is inaccurately indicated in line 97”

This was corrected in the manuscript

“Fig. 5c, larger font should be used for the key. Also, the representative genes shown at the edge of the heatmap are somewhat misleading.”

Font size was changed and representative genes were arranged with brackets as to not to imply that they were associated with a specific line on the heatmap, but rather indicate that they were members of the cluster. This is also indicated in the Figure legend. Many other publications in top tier journals have used this display methodology to bring attention to genes of interest within large datasets, and as we found this compelling, we borrowed that format.

“Fig. 5d is inaccurately indicated in line 168”

This was corrected in the manuscript

“Lines 196-197: it is not clear what the authors mean by “However, we found that these signals can also induce ILC2-10”?”

We thank the reviewer for catching this typo from an earlier draft, and the sentence has been removed.

Reviewer 2

“Page 2 lines 1-2: “Type-2 innate lymphoid (ILC2) cells share cytokine production and transcription factor expression with CD4+ Th2 cells.”

“Page 2 line 2: “but it is unknown if ILC2 cells can undergo additional functional diversification.” This sentence is confusing: what about plasticity like the inflammatory ILC2 that are able to produce IL-17 when cultured with IL-2 and IL-7 (Huang et al, Nat Immunol 2015) for example?”

We thank the reviewer for this suggestion and we have modified the text to state that the functional diversity of the ILC2 lineage has not been fully explored.

“Page 2 lines 11-12: “These data demonstrate previously unappreciated heterogeneity in ILC2 cell responses.” In line with the previous remark, some heterogeneity in ILC2 responses has already been demonstrated such as IFN- γ production by ILC2 during lung inflammation. A relative play down such as “a previously unappreciated IL-10 producing ILC2 generation” would be appreciated.”

We agree with the reviewer here and have changed the text to match the suggestion. We write that “these data demonstrate the generation of a previously unappreciated IL-10 producing ILC2 cell population.”

“Page 3 lines 15-16: “ILCs, have been identified in mice and humans and have many parallels to CD4+ helper T (Th) effector cell subsets.” It would be more accurate to speak of helper-like ILC rather than ILC OR of T cells in general rather than CD4+ Th because ILC include the cytotoxic NK.”

“Page 3 lines 16-18: “In this regard, Type-1 ILCs (ILC1 cells), Type-2 ILCs (ILC2 cells), and Type-3 ILCs (ILC3 cells) have been compared to Th1, Th2, and Th17 cells, respectively.” The actual nomenclature includes NK cells in the Type-1 ILCs group and LTi in the Type-3 ILCs group.”

We have changed this section of the introduction to read as follows:

“Cells of the innate immune system termed innate lymphoid cells (ILCs), have been identified in mice and humans, and helper-like ILCs have many parallels to CD4+ helper T (Th) effector cell subsets. In this regard, some subsets within the Type-1 ILCs (ILC1 cells), Type-2 ILCs (ILC2 cells), and Type-3 ILCs (ILC3 cells) populations have been compared to Th1, Th2, and Th17 cells, respectively.

“Page 3 lines 20-23: Although this difference in cell generation is important, it should not be undermined that other differences are important as well such as the lack of antigen-specific receptors derived from gene rearrangements dependent on RAG recombinases or the tissue-residency of ILC that are rare in the circulating blood and represent a minority of lymphoid tissue cells.”

We had focused here on the adaptability of T cells vs. ILCs as innate immune mediators, but so as not to overstate we have altered the Introduction.

“Page 3 lines 24-25: “ILC2 cells have a beneficial role in eradication of parasitic helminthes.” A reference is missing there.”

The reference has been added.

Results

“Page 4 lines 52-54: “Genes encoding cell surface molecules used for cell isolation (Ptpcr, Thy1, and Il1r1) or markers of the ILC2 cell type (H2-Ab1, Icos, Il2ra) were expressed similarly in both cell populations.” What about klrg1 and il17rb both surface markers also expressed by ILC2 that are labeled in red in the figure 1e?”

We agree with the reviewer that these are important points. We have changed the text accordingly. It now reads as follows:

“IL-33 mediated ILC2 cell activation led to other significant changes in gene expression (**Fig. 1d**), including upregulation of *Klrg1* and *Mki67*, encoding cell activation and proliferation markers (**Fig. 1e**), and *Il13*, *Il6* and *Arg1* (**Fig. 1g**), involved in proliferation and inflammatory functions of ILC2 cells.”

“Page 6 lines 86-88: What about the difference between IL10+ Th1 and IL10- Th1?”

This is interesting but we did not ourselves produce a dataset to address this, as we did for Treg. However, Maf, AhR, IRF4, ROG and Egr2 are thought to play key roles in Tr1 cells and are not among our differentially expressed genes (Table S2) with the exception of IRF4, which is less highly expressed by ILC2₁₀. ILC2₁₀ cells also do not express mRNA for LAG-3, a key marker of Tr1 cells⁵. This is now mentioned in somewhat abbreviated form in the discussion. We focused on Tregs since they are also found in the lung and can be IL-33 responsive. Since no one has ever found Foxp3-expressing ILCs, there has

been long standing interest in whether an analogous ILC regulatory population exists that uses a transcriptional regulator other than FOXP3. Thus, we thought it of interest that ILC2₁₀ cells are molecularly distinct from Tregs, as well as ILC2_{act}.

Discussion

“Page 11 lines 209-210: The parallels between CD4+ Th effector cells and helper-like ILCs have oft been noted, yet identification of diversified effector populations within the ILC2 lineage has not been reported.”

We appreciate the reviewer’s clarification here and have changed the sentence as suggested.

Figures

“Figure 1b and c: The color code should be changed because blue and red both reminds of all the graphs were differentially expressed genes are labeled in red.”

We have changed the color scheme of Fig. 1b,c appropriately.

“Figure 3a: Normally, the contribution of each variable to all the dimensions of the PCA can be determined. This could be interesting to look at in order to understand what are the variables contributing the most to the 2nd dimension, meaning, what are the variables upon which there is the difference between ILC233, ILC2act and ILC210 that are not differentiated by the 1st dimension.”

We thank the reviewer for this suggestion, and we subsequently analyzed the data in this way. However, as there were such a large number of genes comprising each PCA dimension, any individual gene was a minor contributor. Thus, we didn’t find the data as informative as identifying differentially expressed genes as defined in the paper.

“Figure 3f: In line with the previous remark, the color code is confusing because the ILC2act are in blue although they were represented as green dots in the principal component analysis graphs Figure 3a and b.”

We thank the reviewer for bringing this to our attention. We have changed the color scheme as suggested.

“Figure 4b,c,d and e: It could be informative to look at the amount of IL-10 produced by ILC2, ILC210 with the same design of experiment to see if the cells are more potent of IL-10 production when recalled 14 or 30 days after the first induction.”

While we did not directly compare the amount of IL-10 the cells produce, we did sort purify IL-33 re-challenged reporter positive and reporter negative ILC2 cells at day 15. Similar to day 4, *Il10* mRNA was enriched in the reporter positive cells at day 15 (**Supplementary Fig. 3b**). And in terms of reporter we do not see much difference from day 4 to 15 (unlike addition of IL-2c, which does greatly increase reporter expression (**Fig. 6d,e**).

“Figure 5d: Same remark as previously, it would be interesting to know what are the variables contributing the most to the 1st and 2nd dimension of the PCA, meaning, the variable differentiating the populations represented.”

As above, we reanalyzed the data and as any individual gene was only a very small component of the PCA dimension, the DE analysis was more informative.

Methods

“Antibodies and flow cytometry: somewhere should be ‘specified’ what antibodies the lineage cocktail includes.”

We have corrected this oversight. ILC2 were isolated using the following lineage markers: CD8 α , CD4, CD3, $\gamma\delta$ TCR, CD11b, CD19, B220, Gr-1, CD11c, NK1.1, NKp46, Ter-119, and Fc ϵ RII. This information was added to the methods section with antibody clones included.

1. Seehus CR, *et al.* The development of innate lymphoid cells requires TOX-dependent generation of a common innate lymphoid cell progenitor. *Nature immunology* **16**, 599-608 (2015).
2. Halim TY, *et al.* Group 2 innate lymphoid cells license dendritic cells to potentiate memory TH2 cell responses. *Nature immunology* **17**, 57-64 (2016).
3. Schwanhausser B, *et al.* Global quantification of mammalian gene expression control. *Nature* **473**, 337-342 (2011).
4. Moro K, *et al.* Interferon and IL-27 antagonize the function of group 2 innate lymphoid cells and type 2 innate immune responses. *Nature immunology* **17**, 76-86 (2016).
5. Gagliani N, *et al.* Coexpression of CD49b and LAG-3 identifies human and mouse T regulatory type 1 cells. *Nature Medicine* **19**, 739-746 (2013).

Reviewers' comments:

Reviewer #2 (Remarks to the Author):

Response of the reviewer 2:

Corrections that were made and the addition of experiment using papain as an allergen to induce ILC210 do result in a greater depth of the study.

However, I do have a few points to add:

-Please pay attention to your bi-exponential set-ups in all your FACS plots such as the figure 6c IL-33+ ILC2c to cite one, where there is a lot of cells on the axis.

-Page 5 lines 96-98: Numbers of tissue eosinophils or KLRG1 expression by ILC2 cells should be plotted in the figure.

-Page 12 lines 247: « due to due to »

-Page 12 lines 257-258: « The parallels between CD4+Th effector cells and helper-like ILCs have often been noted, yet the generation of a specific IL-10 expressing effector population within the ILC2 lineage has not been reported.

Reviewer #3 (Remarks to the Author):

The paper by Seehus et al. describes a previously unknown ability of murine ILC2s to produce IL-10 in addition to the type 2 cytokine IL-13. An IL-10 producing population arises after in vivo administration of IL-33 and can be generated in vitro by culture with IL-2, IL-7, IL-33 and RA.

Comments:

- In most figures the ILC2/ILC210 frequencies are plotted, in this case it's important to mention how these frequencies are calculated (percentage of CD45+ lymphocytes or?)
- Following the previous question, in figure 6 a/b the reported frequencies of ILC2s and ILC210s are similar. Does this mean that all ILC2s after treatment with IL-33 and IL2c are IL-10 producing cells?
- Even though in figure 3g it is shown that the expression of type 2 cytokines such as IL-4, IL-5 and IL-13 does not differ between ILC2 and ILC210 cells, it would be very informative to compare the amount of cytokines that these two subsets produce ex vivo. Some data is shown in figure S4c and here it seems that ILC210 cells are more potent producers of IL5 and IL-13, although only one example is shown and no statistics were done. This data could be combined with the in vitro data in figure 5b as here sorted total ILC2s from IL-33 treated animals are used. It would be more informative to compare ILC2 and ILC210 cells from the same animals.
- A similar question arises with regards to the in vitro studies. Does the stimulation with IL-2,7,33 and RA induce cells that produce both IL-13 and IL-10 (Fig 5a)? Also it is known that ILC2s more readily produce IL-13 and that IL-5 is more specific for the type 2 response. Do these cells also remain IL-5 producers?
- The RNA-seq data comparing ILC2s stimulated under basal versus IL-10 polarizing conditions (Fig 5c) reveal that certain surface markers such as IL17RB and ICOS are specific for each subset. Did the authors verify this by flow cytometry? It would strengthen the paper if a surface marker could be identified to identify this population.
- The authors speculate that mast cells and CD103+ DCs are the source of IL-2 and RA respectively and show a modest decrease in IL-10 expression in ILC2s upon IL-33 administration

in mice that lacked either mast cells or CD103+ DCs (Fig 6j). Is the total amount of ILC210 cells in these animals affected? And can the authors show that in wild type animals repeated IL-33 administration indeed induces IL-2 production by mast cells and RA by CD103+ DCs?

- The relevance of the IL-10 production by ILC2s upon (repeated) activation could be a self limiting process to avoid uncontrolled activation. In this respect it is valuable data that upon enhancing the IL-10 production by co-administration of IL-33 and IL2c the eosinophilia is limited. This does raise the question whether the pulmonary inflammation is also attenuated (histology). Furthermore, the mechanism remains unclear as known eosinophil recruiting factors are unaltered or increased. Is it possible that IL-10 affects the eosinophil recruitment or activation directly? This should be addressed.
- On page 6, line 98 the authors should refer to the supplemental figure S1.

R2

Corrections that were made and the addition of experiment using papain as an allergen to induce ILC210 do result in a greater depth of the study.

However, I do have a few points to add:

-Please pay attention to your bi-exponential set-ups in all you FACS plot such as the figure 6c IL-33+ ILC2c to cite one, where there is a lot of cells on the axis.

In response, we have altered the display in Fig. 6c, and reanalyzed all the quantitative data as appropriate to match. In doing so, we also noted that although Fig. 6 and Fig. 7 report entirely independent experiments, the representative FACS plot in the former Fig. 6f was repetitious to that shown in Fig. 7a. Thus, we removed the representative plot in the former Fig. 6f, while maintaining the quantitative data (the new Fig 6f, reanalyzed as in Fig. 6c) that corresponds to the RNA expression analysis (new Fig. 6g).

Page 5 lines 96-98: Numbers of tissue eosinophils or KLRG1 expression by ILC2 cells should be plotted in the figure.

We inadvertently left off the reference to Fig. S1, where this correlative data is shown, and have now corrected this.

-Page 12 lines 247: « due to due to »

We thank the reviewer for catching this typo.

Page 12 lines 257-258: « The parallels between CD4+Th effector cells and helper-like ILCs have oft been noted, yet the generation of a specific IL-10 expressing effector population within the ILC2 lineage has not been reported.

This sentence has been corrected as suggested.

R3

- In most figures the ILC2/ILC210 frequencies are plotted, in this case it's important to mention how these frequencies are calculated (percentage of CD45+ lymphocytes or?)

- Following the previous question, in figure 6 a/b the reported frequencies of ILC2s and ILC210s are similar. Does this mean that all ILC2s after treatment with IL-33 and IL2c are IL-10 producing cells?

We thank the reviewer for pointing out that this might be unclear to the reader. The quantitation reflects a compilation of the data as shown in the representative FACS plots. Thus, the frequency of ILC2 cells is calculated as a percentage of CD45⁺Thy1.2⁺ cells, and the frequency of ILC2₁₀ cells (Thy-1.1⁺) is calculated as the percentage of the total ILC2 cell population. Numbers of cells are for the total lung. This has been clarified in the Figure legends.

- Even though in figure 3g it is shown that the expression of type 2 cytokines such as IL-4, IL-5 and IL-13 does not differ between ILC2 and ILC210 cells, it would be very informative to compare the amount of cytokines that these two subsets produce ex vivo. Some data is shown in figure S4c and here it seems that ILC210 cells are more potent producers of IL5 and IL-13, although only one example is shown and no statistics were done. This data could be combined with the in vitro data in figure 5b as here sorted total ILC2s from IL-33 treated animals are used. It would be more informative to compare ILC2 and ILC210 cells from the same animals.

- A similar question arises with regards to the in vitro studies. Does the stimulation with IL-2,7,33 and RA induce cells that produce both IL-13 and IL-10 (Fig 5a)? Also it is known that ILC2s more readily produce IL-13 and that IL-5 is more specific for the type 2 response. Do these cells also remain IL-5 producers?

Together, the RNA-seq data (Fig. 3g), and cellular data in Fig. 5b, and Fig. S4c, indicate that ILC2₁₀ make IL-5 and IL-13. Of the 3, only the RNA-seq approach is directly ex vivo without additional stimulation (Fig. 5b is 6 days of culture with continued stimulation, and Fig. S4c is PMA/ionomycin stimulation, which reveals cytokine production potential).

More to the point, we fail to see how this will change the interpretation of the data in the absence of biology indicating that a quantitative difference in cytokine production is important. The importance of Fig. S4c along with the RNA-seq dataset, however, is that it demonstrates that reduced production of IL-5 and IL-13 by ILC2 cells is not a simple explanation for the decrease in eosinophilia associated with expansion of ILC2₁₀ cells by IL-2c.

- The RNA-seq data comparing ILC2s stimulated under basal versus IL-10 polarizing conditions (Fig 5c) reveal that certain surface markers such as IL17RB and ICOS are specific for each subset. Did the authors verify this by flow cytometry? It would strengthen the paper if a surface marker could be identified to identify this population.

We agree with the reviewer that having cell surface markers that specifically identify the ILC2₁₀ subset would be of great utility. We explored this by identifying potential markers that came from our RNAseq datasets, as this dataset is reflective of in vivo phenotypes (i.e. no in vitro stimulation or long-term culture). Unfortunately, we have not found a surface marker that adequately identifies the ILC2₁₀ cells with the reagents we tested. This speaks to the difficulty of identifying effector states solely by cell surface markers, much like Th effector cell subsets that rely most accurately on specific cytokine expression profiles for identification. We also pointed out in the paper that there is incomplete overlap in gene expression from in vitro and in vivo generated cells. This too is not unexpected, given the complex nature of the in vivo microenvironment, that only single time points were analyzed, and the likely intricate relationship between cell cycling and cell differentiation as observed for Th effector cells¹. But it is notable that the in vitro work was predictive of the effect of IL-2c stimulation on ILC2₁₀ cell generation. As such cytokine complexes are being tested clinically, this is significant.

- The authors speculate that mast cells and CD103⁺ DCs are the source of IL-2 and RA respectively and show a modest decrease in IL-10 expression in ILC2s upon IL-33 administration in mice that lacked either mast cells or CD103⁺ DCs (Fig 6j). Is the total amount of ILC2₁₀ cells in these animals affected? And can the authors show that in wild type animals repeated IL-33 administration indeed induces IL-2 production by mast cells and RA by CD103⁺ DCs?

To identify ILC2₁₀ cells in this context would require extensive breeding, and likely of both mutations, onto the 10BiT reporter background, since each mutation alone had a modest and partial effect on Il10 mRNA production. As this would require many months and at least two generations of breeding, we think this is outside the scope of the current study. RA production by CD103⁺ DC and IL-2 production by mast cells has been previously reported, although not the only source of these mediators. However, given the modest effect of removing either cell type on IL-10 production by ILC2 cells, the data demonstrate that neither cell type is essential to support generation of these effector cells.

- The relevance of the IL-10 production by ILC2s upon (repeated) activation could be a self limiting process to avoid uncontrolled activation. In this respect it is valuable data that upon enhancing the IL-10 production by co-administration of IL-33 and IL2c the eosinophilia is limited. This does raise the question whether the pulmonary inflammation is also attenuated (histology). Furthermore, the mechanism remains unclear as known eosinophil recruiting factors are unaltered or increased. Is it possible that IL-10 affects the eosinophil recruitment or activation directly? This should be addressed.

IL-10 has been reported to inhibit eosinophilia^{2, 3, 4}, although the role of this cytokine in airway hypersensitivity is less certain and may depend on the specific system. In addition, there is data that some eosinophil populations poorly

express RNA for the IL-10 receptor⁵, and thus it is not known if IL-10 can directly impact eosinophil function. We have added this to the Discussion. The data demonstrate that ILC2₁₀ cells are molecularly distinct from the bulk population of activated ILC2 cells, and the biological role of ILC2₁₀ cells is likely to be complex, and not solely due to IL-10 production. We focused on RAG deficient mice and eosinophilia to specifically eliminate any confounding aspects of inflammation caused by the adaptive immune system and to focus on one known impact of ILC2 cell activation and cytokine production (as also evidenced by the lack of eosinophilia in NSG mice, Fig. 7).

• On page 6, line 98 the authors should refer to the supplemental figure S1.

We thank the reviewer for pointing out this oversight.

Finally, we have updated the Discussion based on recently published data since submission of this study, concerning regulatory populations of ILC, but which are not ILC2 lineage cells as we have described.

1. Proserpio V, *et al.* Single-cell analysis of CD4+ T-cell differentiation reveals three major cell states and progressive acceleration of proliferation. *Genome Biol* **17**, 103 (2016).
2. van Scott MR, Justice JP, Bradfield JF, Enright E, Sigounas A, Sur S. IL-10 reduces Th2 cytokine production and eosinophilia but augments airway reactivity in allergic mice. *Am J Physiol Lung Cell Mol Physiol* **278**, L667-674 (2000).
3. Zuany-Amorim C, *et al.* Interleukin-10 inhibits antigen-induced cellular recruitment into the airways of sensitized mice. *The Journal of clinical investigation* **95**, 2644-2651 (1995).
4. Grunig G, Corry DB, Leach MW, Seymour BW, Kurup VP, Rennick DM. Interleukin-10 is a natural suppressor of cytokine production and inflammation in a murine model of allergic bronchopulmonary aspergillosis. *Journal of Experimental Immunology* **185**, 1089-1099 (1997).
5. Zigmund E, *et al.* Macrophage-restricted interleukin-10 receptor deficiency, but not IL-10 deficiency, causes severe spontaneous colitis. *Immunity* **40**, 720-733 (2014).

REVIEWERS' COMMENTS:

Reviewer #3 (Remarks to the Author):

no further comments